# A systematic review and meta-analysis of the diagnostic accuracy after preimplantation genetic testing for aneuploidy

Vanessa Bacal[1,2]*, Angela Li[1], Heather Shapiro[1,2], Urvi Rana[3], Rhonda Zwingerman[4], Lisa Avery[5,6], Alina Palermo[2], Eleni Philipoppolous[7], Crystal Chan[1,8]

1 Department of Obstetrics and Gynaecology, University of Toronto, Canada, 2 Mount Sinai Fertility, Mount Sinai Hospital, Toronto, Canada, 3 Department of Obstetrics and Gynecology, Henry Ford Macomb Hospital, Clinton Township, United States of America, 4 Twig Fertility, Toronto, Canada, 5 Biostatistics Research Unit, University Health Network, Toronto, Canada, 6 Department of Biostatistics, Dalla Lana School of Public Health, University of Toronto, Canada, 7 McGill University, Montreal, Canada, 8 Markham Fertility Centre, Markham, Canada

* Vanessa.bacal@sinaihealth.ca

## Abstract

### Objective

Aneuploidy accounts for many pregnancy failures and congenital anomalies. Pre-implantation genetic testing for aneuploidy (PGT-A) is a screening test applied to embryos created from in vitro fertilization to diminish the chance of an aneuploid conception. The rate of misdiagnosis for both false aneuploidy (false positive) and false euploidy (false negative) test results is unknown. The objective of this study was to determine the rate of misclassification of both aneuploidy and euploidy after PGT-A.

### Data sources

We conducted a systematic review and meta-analysis. We searched Medline, Embase, Cochrane Central, CINAHL and WHO Clinical Trials Registry from inception until April 10, 2024. The protocol was registered in International Prospective Register of Systematic Reviews (PROSPERO CRD 42020219074).

### Methods of study selection

We included studies that conducted either a pre-clinical validation of the genetic platform for PGT-A using a cell line, studies that compared the embryo biopsy results to those from the whole dissected embryo or its inner cell mass (WE/ICM), and studies that compared the biopsy results to prenatal or postnatal genetic testing.

### Tabulation, Integration, and Results

Two independent reviewers extracted true and false positives and negatives comparing biopsy results to the reference standard (known karyotype, WE/ICM, pregnancy

**Data availability statement:** All relevant data are within the manuscript and its Supporting Information files.

**Funding:** Our project was supported by funding from the Department of Obstetrics and Gynaecology at Sinai Health, Toronto, Ontario, Canada. There was no additional external funding received for this study. The funders had no role in study design, data collection and analysis, decision to publish, or preparation of the manuscript.

**Competing interests:** The authors have declared that no competing interests exist.

outcome). For preclinical studies, the main outcome was the positive and negative predictive values. Misdiagnosis rate was the outcome for pregnancy outcome studies. The electronic search yielded 6674 citations, of which 109 were included. For WE/ICM studies (n=40), PPV was 89.2% (95% CI 83.1-94.0) and NPV was 94.2% (95% CI 91.1-96.7, $I^2$=42%) for aneuploid and euploid embryos, respectively. The PPV for mosaic embryos of either a confirmatory mosaic or aneuploid result was 52.8% (95% CI 37.9-67.5). For pregnancy outcome studies (n=43), the misdiagnosis rate after euploid embryo transfer was 0.2% (95% CI 0.0-0.7%, $I^2$=65%). However, the rate for mosaic transfer, with a confirmatory euploid pregnancy outcome, was 21.7% (95% CI: 9.6-36.9, $I^2$=95%).

## Conclusion

The accuracy of an aneuploid result from PGT-A is excellent and can be relied upon as a screening tool for embryos to avoid aneuploid pregnancies. Similarly, the misdiagnosis rate after euploid embryo transfer is less than 1%. However, there is a significant limitation in the accuracy of mosaic embryos.

## Introduction

Aneuploidy accounts for most miscarriages, as well as congenital anomalies and implantation failure in women [1]. Preimplantation genetic testing for aneuploidy (PGT-A) is a screening test that is applied to embryos created by in vitro fertilization (IVF). Using PGT-A should optimize implantation and live birth rates per embryo transfer, and decrease miscarriage rates [2]. In current practice, embryos are cultured to the blastocyst stage, at which point five to ten cells are biopsied from the embryos and DNA is extracted to perform genetic testing. Embryos that screen negative (i.e., euploid or chromosomally normal) are preferentially selected for transfer. Those that screen positive (i.e., aneuploid or chromosomally abnormally) are not selected for transfer, and patients are usually advised to discard them. A third category of embryos are those that screen mosaic, which is defined as a combination of both euploid and aneuploid cells [3]. Because the reproductive potential and implications of transferring these embryos are complex, they are prioritised below a euploid embryo.

False negative results occur and lead to ongoing aneuploid pregnancies and aneuploid pregnancy losses [4,5]. In vitro studies have also demonstrated false positive PGT-A results, as embryos initially screened as aneuploid have been demonstrated to be euploid upon resampling [6,7]. Sources of error with PGT-A include human factors (misinterpretation of results, transfer of the wrong embryo), technical factors (DNA contamination, screening platform utilized, biopsy technique), intrinsic sampling error (e.g., from embryo mosaicism), and the chance of spontaneous conception around the time of transfer [8,9].

To validate PGT-A as a selection tool for ET, and to aid in patient counselling, the risk of an aneuploid pregnancy after euploid ET, as well as the risk of euploidy after initial aneuploidy classification should be known.

The first objective of this systematic review and meta-analysis was to assess the false negative rate of PGT-A (where embryos screened as euploid are actually aneuploid), and to estimate the chance of error with PGT-A using clinical studies on aneuploid conceptions after euploid embryo transfers. The second objective was to assess the false positive rate of PGT-A (where embryos initially screened as aneuploid are actually euploid) using studies that evaluated pregnancy outcomes after aneuploid embryo transfer, clinical non-selection studies, and *in vitro* studies that involved resampling of embryos.

## Materials and methods

### Eligibility criteria

For the first objective, we included studies that reported on outcomes of patients who underwent PGT-A with subsequent embryo transfer. We assessed studies that performed genetic testing of ongoing pregnancies (via amniocentesis or genetic testing of infants), genetic testing of products of conception after euploid ET with subsequent pregnancy loss, or physical examination of infants. While most aneuploid pregnancies will not continue to term and most aneuploid individuals have clear phenotypic abnormalities, not all do (e.g., XXX, XXY), nor would all children with true chromosomal mosaicism. We therefore excluded studies that only described "healthy infants" without a description of either genetic analysis or examination findings to verify the initial PGT-A screening diagnosis.

For the second objective, we included studies evaluating embryos with an aneuploid or mosaic result after PGT A, that were either rebiopsied, had genetic testing of pregnancy, pregnancy loss. This group also included embryos in a research setting that underwent TE biopsy with comparison to either its ICM or whole dissected embryo (WE). However, where studies only rebiopsied the TE, Finally, we included preclinical studies of cell lines with known karyotypes that evaluated the efficiency and reliability of the PGT-A testing platform.

We included case series, (with three or more patients), case control studies, cohort studies, non-selection studies, and randomized controlled trials. We included all studies with sufficient detail in their validation for replication. As we anticipated that many studies, particularly the pre-clinical designs, would demonstrate high validity of the testing platform and would not proceed to publication, we elected to include abstracts if full length manuscripts were not available, provided there was sufficient information for narrative review and/or a two-by-two table. There were no restrictions by type of setting, or length of follow-up, however, we only included studies reported in English and French. Studies were excluded if the reference test was a screening test rather than a diagnostic test (for example, non-invasive prenatal screening), case reports, case series with fewer than three patients, or studies validating either PGT for monogenic disorders (PGT-M) or for structural rearrangements (PGT-SR) only without a concurrent PGT-A analysis. Studies that used fluorescence in situ hybridization (FISH) either as the index test or reference standard were excluded, because it has been replaced by 24 chromosome analysis (comprehensive chromosome screening, CCS) [10,11].

### Outcomes

The outcomes of interest included positive predictive value of an aneuploid or mosaic embryo, and the negative predictive value of a euploid embryo. For pregnancy outcomes, we considered prenatal diagnosis, products of conception, neonatal testing or examination as the reference standard. For rebiopsy studies, we considered the ICM or WE as the reference standard. For the preclinical studies, we considered the known karyotype of the cell lines to be the reference standard.

### Information sources

We developed a comprehensive search strategy with relevant keywords and MeSH terms with the guidance of an information specialist, tailored to Medline and applied to Embase, Cochrane Central, WHO Clinical Trials Registry and ClinicalTrials.gov from inception until April 10, 2024. Specific keywords include preimplantation genetic testing, aneuploidy, chromosomal aberrations, false negative and false positive, sensitivity, specificity, predictive value, validity (S1 File). We

imported and managed studies in Covidence systematic review software (Veritas Health Innovation, Melbourne, Australia; available at www.covidence.org).

### Screening

The screening was performed in two stages, initially with title and abstract followed by evaluation of full texts, by two independent reviewers (VB and UR/AL) according to the eligibility criteria. We resolved disagreements by consensus. Where consensus could not be reached, the final decision was made by a senior author (CC). Reasons for exclusion were documented (S2 File).

### Data extraction

We extracted all relevant information from studies that meet final inclusion criteria including study design, sample size, primary outcome, PGT-A platform used, transfer of one or two embryos, completeness of follow up, type of POC testing (karyotype vs array), type of testing of ongoing pregnancies (chorionic villus sampling (CVS), amniocentesis, cord blood, physical exam of newborn), and key study findings (estimation of error rate). Two independent reviewers (VB and AL) extracted and compared the data in duplicate from the selected studies. We resolved discrepancies by consensus.

### Data analysis

Quality of individual studies was determined using the QUADAS-2 tool for diagnostic accuracy [12]. Where data could be synthesized quantitatively, we performed a meta-analysis using a random effects model with the meta package in R software version 4.2.1 [13]. The specific outcomes we analyzed included false negative, false positive, negative predictive value and positive predictive value, with the reference test as the genetic testing from an ongoing pregnancy or the infant, or the resampled embryo in the event of the initial aneuploidy diagnosis. Where data were missing, only selected measures of diagnostic accuracy were calculated provided sufficient information was available. If insufficient information to estimate any measures of diagnostic accuracy, a narrative review only was conducted. We have reported $I^2$, a measure of heterogeneity across studies where values > 75% indicate high variability across study results. To contextualise the heterogeneity, we have also computed prediction intervals, which indicate the range of effect sizes we would expect to see in a new study. Wide prediction intervals indicate high uncertainty in future results. We performed subgroup analyses evaluating the impact of cleavage stage biopsy (day three) vs blastocyst biopsy (day five to seven), and genetic platform used to perform PGT-A (next generation sequencing (NGS), array comparative genomic hybridization (aCGH), single-nucleotide polymorphisms (SNP) microarray, or polymerase chain reaction (PCR)), and publication type. For pregnancy outcomes, the denominator is incalculable due to inability to validate each embryo transferred either due to failed implantation, early pregnancy loss, or failure to obtain a DNA sample from the pregnancy or infant. We therefore reported the misdiagnosis rate, defined by the number of false negatives (for euploid ET) or false positives (for aneuploid or mosaic ET) divided by total number of embryos transferred, as previously described [4].

The protocol was registered in International Prospective Register of Systematic Reviews (PROSPERO CRD 42020219074) and was reported according to Preferred Reporting Items for a Systematic Review and Meta-analysis of Diagnostic Test Accuracy Studies (PRISMA-DTA) statement [14].

## Results

The electronic search yielded 6674 citations, of which 109 met the inclusion criteria (Fig 1). We included 19 pre-clinical validation studies [15–37], 40 that compared the TE biopsy to the ICM or WE [6,7,24,25,38–73], and 56 clinical studies that investigated pregnancy outcomes after ET [4,5,7,16,27,57,74–123]. Studies that conducted a mixed design (i.e., cell line pre-clinical validation and pregnancy outcomes) were extracted and meta-analyzed separately in their appropriate category. Preclinical studies are described in S3 File. Results are available in S1–S3 Tables and S1–S3 Fig.

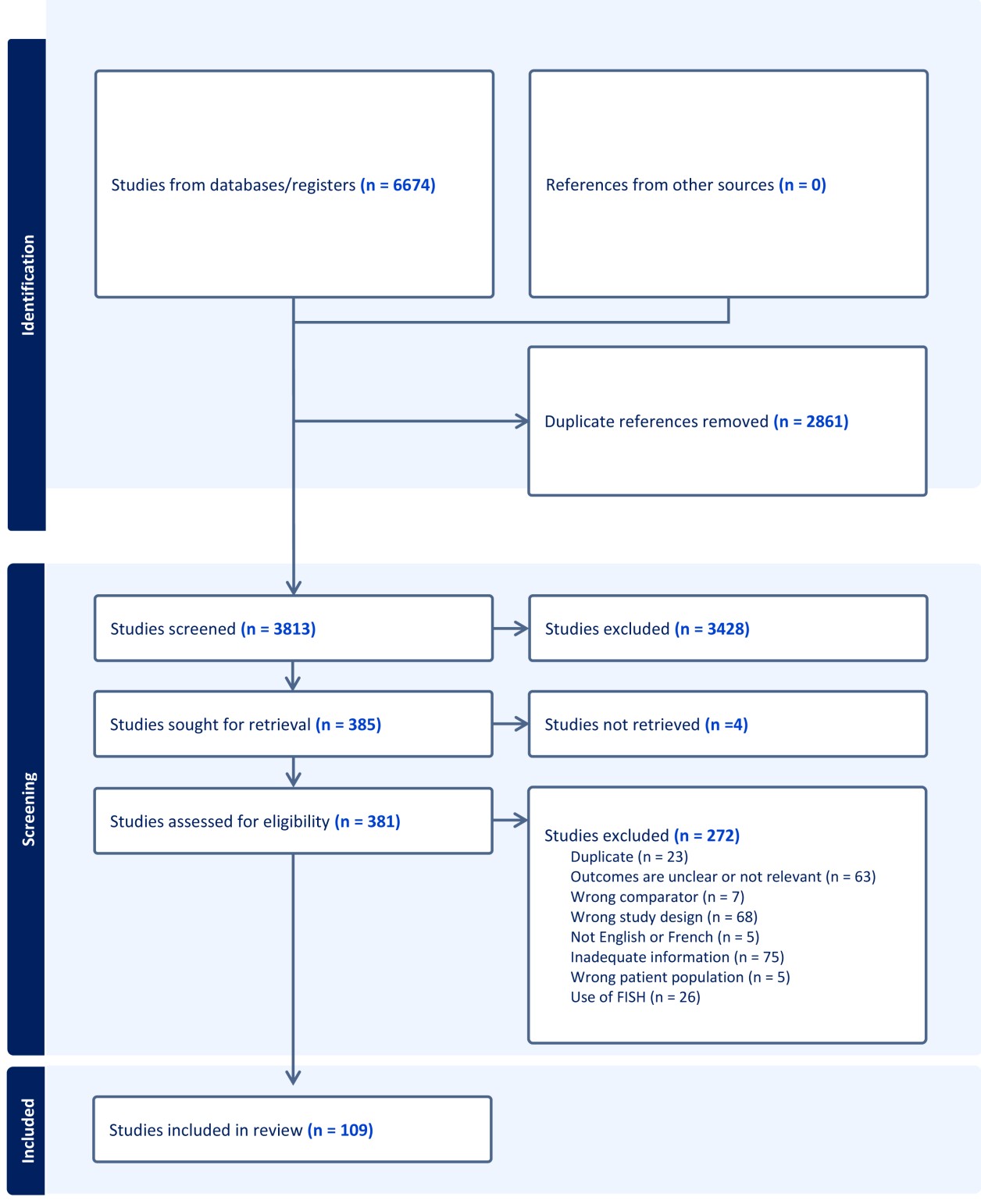

**Fig 1. PRISMA flow diagram.**

## Whole embryo or inner cell mass

We included 40 studies in the meta-analysis (Table 1). Two studies evaluated cleavage stage results (two evaluated at both cleavage and blastocyst stage), and the remaining studies evaluated blastocyst stage embryos (trophectoderm biopsy). The point estimate of the NPV was 94.2% (95% CI 91.1–96.7, $I^2 = 42\%$) with a prediction interval of 86.2 to 98.9. This means that that the probability of an embryo being euploid if there was a negative test result is > 86% (Fig 2a). The PPV of an aneuploid result was 89.2% (95% CI 83.1–94.0) with a wide prediction interval (42.0–100.0) (Fig 2b). Nine studies evaluated the concordance among mosaic screened embryos, [25,45,56,57,62–64,68,70], which was typically defined by percentage of aneuploid cells (20–80%) (Table 2). The overall PPV for mosaic embryos was 52.8% (95% CI 37.9, 67.5) with a wide prediction interval of 11.7 to 91.7 (Fig 2c). Other measures of diagnostic accuracy are presented in S4 Fig. Two-by-two table is available in S4–S5 Tables.

We performed sensitivity analyses investigating the impact of PGT-A platforms (NGS vs other platforms), reference comparators of ICM biopsy or WE, and publication of results (conference abstract versus full text). There were no differences comparing NGS to other platforms or full-text publications compared to conference abstracts (S5, S6 Fig).

Measures of diagnostic accuracy were slightly higher when comparing ICM biopsy to the WE with less heterogeneity (S7 Fig). While the impact of stage of biopsy revealed a higher overall accuracy at blastocyst compared to cleavage embryos (84.3 vs 60.0), the heterogeneity was still high (> 80%) (S8 Fig).

The quality assessment for studies that performed CCS analysis was largely either unclear or low concern for risk of bias (S6 Table). Most studies evaluated donated embryos that were initially diagnosed as aneuploid or mosaic, or of poor quality that were unsuitable for transfer contributing to partial verification bias. Overall, there was low concern that the interpretation of the index test differed from the review question.

## Studies reporting pregnancy outcomes after PGT-A and ET

There were 56 studies that evaluated pregnancy outcomes after transfer of a PGT-A and ET (Table 3). Of these, 26 exclusively used NGS, and 23 used another platform [86,114].

## Euploid embryo transfers

In the 42 studies that evaluated outcomes of euploid ETs, validation was performed against one or more of: amniocentesis, CVS, products of conception (POC), postnatal genetic testing, or complete physical examination (Table 3). There were 10,641 reported pregnancies among 20,196 embryos transferred. Of these pregnancies, only 1367 validated the genetic status against one of the above methods. Overall, the misdiagnosis rate for a false negative upon euploid ET was 0.2% (95% CI 0.0–0.7%, $I^2 = 65\%$), with a prediction interval from 0–3.4% (Fig 3a). Among tested POC, there were 22 euploid embryos that were diagnosed with a microdeletion below the limit of detection of PGT-A; as this was considered an incidental finding rather than a misdiagnosis, we classified this as euploid. Two-by-two table is available in S7 Table.

## Aneuploid embryo transfers

Clinical studies on this population of embryos have been limited in sample size due ethical limitations. There were two studies that described 30 cases of aneuploid ET resulting in 15 pregnancies, of which five led to healthy live births [7,75]. There were three non-selection studies included in our search [97,98,103], of which two reported the number of embryos transferred [97,103]. The misdiagnosis rate upon aneuploid-screened ETs (including the non-selection studies) was 10.6% (95% CI:0–38% $I^2 = 91\%$) (Fig 3b). The misdiagnosis rate for a false positive among aneuploid-screened embryos in the two non-selection studies was 0.4 (95% CI: 0.0–0.3%, $I^2 = 91\%$) (Fig 3c). The non-selection study by Scott (2012) revealed that of 99 transferred embryos that would have been classified as aneuploid, four resulted in live births [97]. Of these four

Table 1. Characteristics of whole embryo or ICM studies.

| Study | Publication type | Country | Number of patients | Number of embryos tested | Patient population | Mean female age (years) | Stage of embryo development at biopsy | Initial diagnosis | Index test: Method of aneuploidy detection (initial TE biopsy) | Reference standard: Method of aneuploidy detection | Reference standard |
|---|---|---|---|---|---|---|---|---|---|---|---|
| Brezina PR 2012[38] | Conference abstract | USA, China | 22 | 228 | Patients who underwent IVF with PGT-A for recurrent pregnancy loss | Not described | Cleavage stage (Day 3) | Aneuploid | SNP microarray | SNP microarray | ICM biopsy |
| Chavli EA 2022[39] | Full text | China | 12 | Not described | Patients who underwent PGT-A±PGT-SR | 30.8; Range: 24–43 | Blastocyst (Day 5–6) | Aneuploid or mosaic | NGS | NGS | ICM biopsy |
| Chen J 2021[40] | Full text | China | Not described | 265 | Not described | Range: 24–39 | Blastocyst (Day 5–6) | Undiagnosed | NGS | NGS | Whole embryo |
| Chen L 2021[41] | Full text | Taiwan | 12 | Not described | Patients who underwent IVF/ICSI for infertility who had achieved a successful live birth and donated unused blastocysts. | 34.4; Range 26–43 | Blastocyst (Day 5) | Undiagnosed | NGS | NGS | ICM biopsy |
| Chuang T-H 2018[42] | Conference abstract | Brazil | Not described | Not described | Not described | Not described | Blastocyst (Day 5) | Aneuploid | Not described | NGS | Whole embryo |
| Franco JG 2023[43] | Conference abstract | USA | Not described | Not described | Not described | Not described | Blastocyst | Aneuploid or euploid | Not described | NGS | Whole embryo |
| Friedenthal J 2021[44] | Conference abstract | USA | Not described | Not described | Not described | Not described | Blastocyst | Aneuploid or mosaic | NGS | NGS | ICM biopsy |
| Garrisi G 2016[45] | Full text | Italy | Not described | 8137 | Patients who underwent IVF with PGT-A for infertility | 38 (±3); Range: 31–44 | Blastocyst | Segmental aneuploid or euploid | NGS | NGS | ICM biopsy |
| Girardi L 2020[46] | Full text | USA | 2 | Not described | Not described | Not described | Blastocyst (confirmed by author) | Aneuploid | NGS | Array CGH | Whole embryo |
| Gleicher N 2016[7] | Full text | Australia | Not described | 14075 | Patients who underwent PGT-A±PGT-M | Not described | Blastocyst (Day 5–6) | Aneuploid | NGS | NGS | ICM biopsy |
| Grkovic S 2022[47] | Full text | China | 13 | Not described | Patients with parental chromosomal rearrangement | Not described | Blastocyst | Unbalanced or developmental arrest | NGS | NGS | ICM biopsy |
| Gui B 2016[48] | Conference abstract | USA | Not described | 34210 | Not described | 36.7 (± 4.2) | Blastocyst | Chromosomal deletions, euploid and aneuploid | NGS | NGS | ICM biopsy |

(Continued)

| Study | Publication type | Country | Number of patients | Number of embryos tested | Patient population | Mean female age (years) | Stage of embryo development at biopsy | Initial diagnosis | Index test: Method of aneuploidy detection (initial TE biopsy) | Reference standard: Method of aneuploidy detection | Reference standard |
|---|---|---|---|---|---|---|---|---|---|---|---|
| Hruba M 2018[49] | Full text | China | 23 | Not described | Patients who underwent PGT-A±PGT-SR for either recurrent pregnancy loss or parental balanced translocations | Range: 24–44 | Blastocyst | Aneuploid or unbalanced | Array CGH | NGS | ICM biopsy |
| Huang J 2017[50] | Full text | USA | 13 | 52 | Patients who underwent IVF with PGT-A for infertility; three patients had a parental balanced translocation | 35 | Blastocyst (Day 5–6) | Undiagnosed | NGS | NGS | Whole embryo |
| Huang L 2019[51] | Conference abstract | Russia | Not described | Not described | Not described | Not described | Blastocyst | Aneuploid | Array CGH | Array CGH | ICM biopsy |
| Kaimonov V 2019[52] | Full text | Canada | 26 | Not described | Not described | 37.5 (± 5.8); Range 25–45 | Blastocyst (Day 5–6) | Aneuploid | NGS | NGS | Whole embryo |
| Kuznyetsov V 2018[53] | Full text | United Arab Emirates | 42 | Not described | Patients who underwent IVF/ICSI with PGT-A for infertility | 33.9; Range: 24–46 | Blastocyst (Day 5–6) | Undiagnosed | NGS | NGS | ICM biopsy |
| Lawrenz B 2019[54] | Conference abstract | USA | 2766 | Not described | Patients who donated surplus embryos | Not described | Blastocyst | Aneuploid, segmental aneuploid or euploid | NGS | NGS | ICM biopsy |
| Lee R 2022[55] | Full text | China | 22 | 3738 | Patients who underwent IVF with PGT-A±PGT-SR. 11 patients had parental karyotype abnormality | Not described | Blastocyst (Day 5–6) | Aneuploid or mosaic | Array CGH (2014–2017); NGS (2018–2019) | NGS | Whole embryo |
| Li X 2021[56] | Full text | Taiwan | 108 | Not described | Patients who underwent IVF with PGT-A for infertility, recurrent pregnancy loss, or combined PGT-M | Not described | Blastocyst (Day 5–6) | Mosaic | NGS | NGS | ICM biopsy |
| Lin P-Y 2020[57] | Full text | China, USA | 51 | 258 | Patients who underwent IVF with PGT-A for either infertility with recurrent pregnancy loss or advanced reproductive age | Not described | Blastocyst (Day 5–6) | Aneuploid | Array CGH | Array CGH | ICM biopsy |
| Liu J 2012[58] | Full text | Spain | 29 | 92 | Patients who underwent IVF with PGT-A for infertility or recurrent pregnancy loss | 41.3 (± 3.4) | Blastocyst (Day 5–6) | Aneuploid | NGS | NGS | ICM biopsy |
| Lledo B 2021[59] | Full text | USA | Not described | Not described | Not described | Not described | Blastocyst | Aneuploid | PCR | PCR | ICM biopsy |

*(Continued)*

Table 1. (Continued)

| Study | Publication type | Country | Number of patients | Number of embryos tested | Patient population | Mean female age (years) | Stage of embryo development at biopsy | Initial diagnosis | Index test: Method of aneuploidy detection (initial TE biopsy) | Reference standard: Method of aneuploidy detection | Reference standard |
|---|---|---|---|---|---|---|---|---|---|---|---|
| Marin D 2017[24] | Full text | Spain | 58 | Not described | Not described | 38.6 (Day 3 cleavage); 36.9 (Day 5 blastocyst) | Cleavage stage (Day 3) or Blastocyst (Day 5) | Aneuploid | Array CGH | Array CGH | Whole embryo |
| McCarty K 2022[60] | Full text | USA | Not described | 89226 | Patients who underwent IVF with PGT-A and donated surplus embryos | 35.5 | Blastocyst | Aneuploid, segmental aneuploid, or euploid | NGS | NGS | Whole embryo |
| Mir P 2016[61] | Full text | Czech Republic | 65 | Not described | Not described | Not described | Blastocyst | Aneuploid or euploid | NGS | NGS | Whole embryo |
| Navratil R 2020[62] | Full text | Israel | 8 | Not described | Patients who underwent IVF with combined PGT-M and PGT-A for monogenic disease | 33.1; Range 26–41 | Blastocyst (Day 5) | Undiagnosed | NGS | NGS | Whole embryo |
| Orvieto R 2016[6] | Full text | China | 18 | Not described | Patients who underwent IVF with combined PGT-SR and PGT-A for parental karyotype abnormality (balanced or Robertsonian translocation, or chromosome inversion) | 31.3 | Blastocyst (Day 5–6) | Abnormal result after PGT-SR | NGS | NGS | Whole embryo |
| Ou Z 2020[63] | Full text | Belgium | 43 | Not described | Not described | 32.2; Range 23–39 | Blastocyst (Day 5–6) | Aneuploid and untested | NGS | NGS | ICM biopsy |
| Popovic M 2018[25] | Full text | The Netherlands, Belgium | 51 | Not described | Not described | 32.8; Range 23–42 | Blastocyst (Day 5) | Aneuploid and untested | NGS | NGS | Whole embryo |
| Popovic M 2019[64] | Full text | 8 different centres in Europe, North America, South America, Asia | 41 | Not described | Patients who underwent IVF with PGT-A for infertility, recurrent pregnancy loss, previous karyotypically abnormal conception, or gender selection | 36.4±5.2; Range 20–44 | Blastocyst (Day 6–7) | Aneuploid | NGS | NGS | ICM biopsy |
| Rubio C 2020[65] | Full text | USA | 17 | Not described | Not described | 39.5±3.3 | Blastocyst (Day 5–7) | Euploid and aneuploid | NGS | NGS for 15 embryos; array CGH for 2 embryos | ICM biopsy |
| Sachdev NM 2020[66] | Full text | Japan | 12 | 20 | Patients who underwent IVF for infertility and donated surplus embryos | 35.6 | Blastocyst (Day 5–6) | Untested | NGS | NGS | Whole embryo |

(Continued)

**Table 1.** (Continued)

| Study | Publication type | Country | Number of patients | Number of embryos tested | Patient population | Mean female age (years) | Stage of embryo development at biopsy | Initial diagnosis | Index test: Method of aneuploidy detection (initial TE biopsy) | Reference standard: Method of aneuploidy detection | Reference standard |
|---|---|---|---|---|---|---|---|---|---|---|---|
| Shitara A 2021[67] | Full text | Japan | 11 | 29 | Patients who underwent IVF for infertility and donated surplus embryos | 34.7±2.7 | Blastocyst (Day 6–7) | Untested | NGS | NGS | Whole embryo |
| Takahashi H 2021[68] | Full text | USA | Not described | 96 | Not described | Not described | Cleavage stage and blastocyst | Undiagnosed | Array CGH | Array CGH | Whole embryo |
| Tobler KJ 2015[69] | Full text | Russia, Estonia | 14 | 16 | Not described | Not described | Blastocyst | Undiagnosed | NGS | NGS | ICM biopsy |
| Tsuiko O 2018[70] | Full text | USA | 45 | Not described | Patients who underwent IVF with PGT-A for infertility | 36.5±5.7 | Blastocyst | Aneuploid | NGS | NGS | ICM biopsy |
| Victor AR 2019[71] | Full text | China | 380 | 1719 | Not described | 31±4.4; Range: 23–44 | Blastocyst (Day 5–6) | Mosaic | NGS | NGS | ICM biopsy |
| Wu L 2021[72] | Full text | China | Not described | Not described | Patients who underwent IVF with PGT-A for infertility | Not described | Blastocyst | Aneuploid | Array CGH | NGS | ICM biopsy |
| Yin B 2021[73] | Full text | China | Not described | Not described | Patients who underwent IVF with PGT-A for infertility and donated surplus embryos | Not described | Blastocyst | Aneuploid | Array CGH | NGS | Whole embryo |

CGH: Comparative genomic hybridization; ICM: Inner cell mass NGS: Next generation sequencing; PGT-A: Preimplantation genetic testing for aneuploidy; PGT-M: Preimplantation genetic testing for monogenic diseases; PGT-SR: Preimplantation genetic testing for structural rearrangements; PCR: Polymerase chain reaction; SNP: Single nucleotide polymorphism.

a. Negative predictive value of euploid embryos

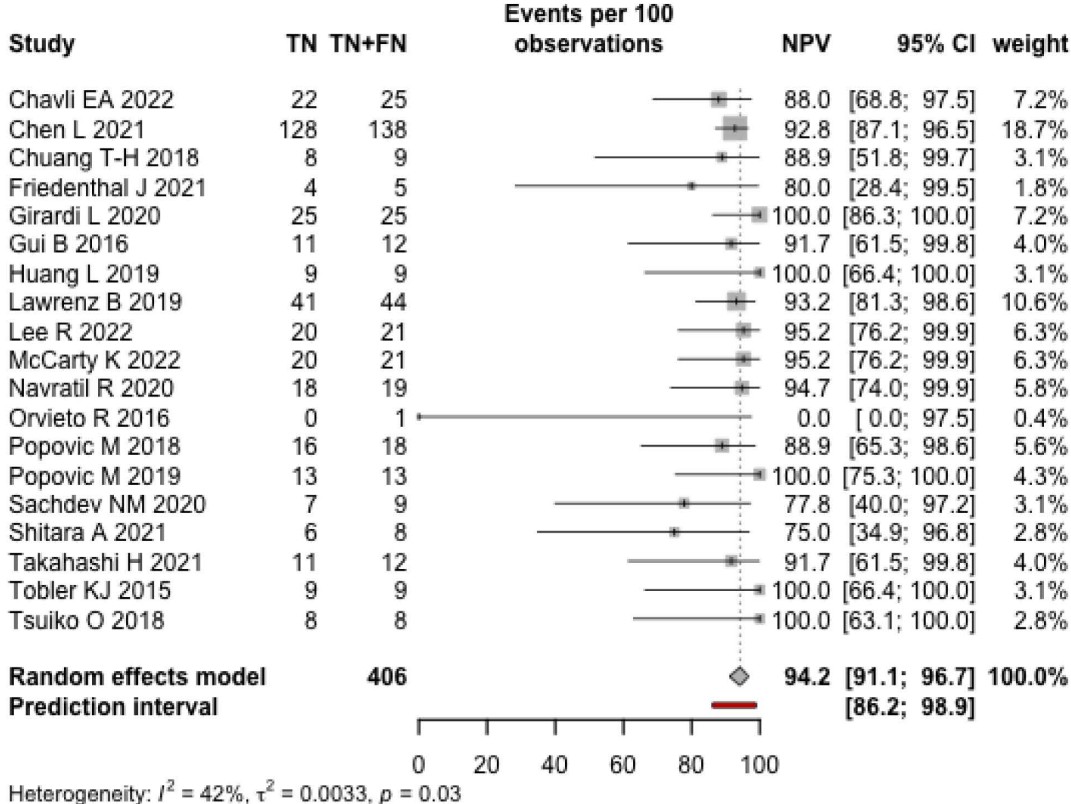

**Fig 2. Forest plots for whole embryo or ICM studies. a. Negative predictive value of euploid embryos. b. Positive predictive value of aneuploid embryos. c. Positive predictive value of mosaic embryos.**

misclassified embryos, three were biopsied at the blastocyst stage, and one was biopsied on day three. In a subsequent trial by Tiegs (2021), 0/102 aneuploid-screened embryos resulted in sustained implantation or delivery (Fig 3c) [103]. Two-by-two tables are available in S8, S9 Tables.

## Mosaic embryo transfers

There were fourteen studies that transferred mosaic embryos, defined variably by either copy number variation or by the proportion of cells that were aneuploid, with ranges from 20–80% or 30–70% being the most common (Table 4) [27,57,74–76,79,80,90,93,94,99,106,116,122]. We classified the outcome as a misdiagnosis when the pregnancy outcome was euploid; however, if the pregnancy was aneuploid or mosaic, this was considered a true positive event, regardless of the number or specific chromosomes involved. Of 2611 embryos transferred, 41 cycles included the transfer of multiple embryos (some of which included euploid embryos). Mosaic ETs resulted in 1157 pregnancies, of which 724 had validated outcome data against which to compare the PGT-A results. The misdiagnosis rate was 21.7% (95% CI: 9.6–36.9, $I^2 = 95\%$) (Fig 3d). Again, two embryos had microdeletions below the limit of detection and were not considered a misdiagnosis. Two-by-two table is available in S10 Table.

b. Positive predictive value of aneuploid embryos

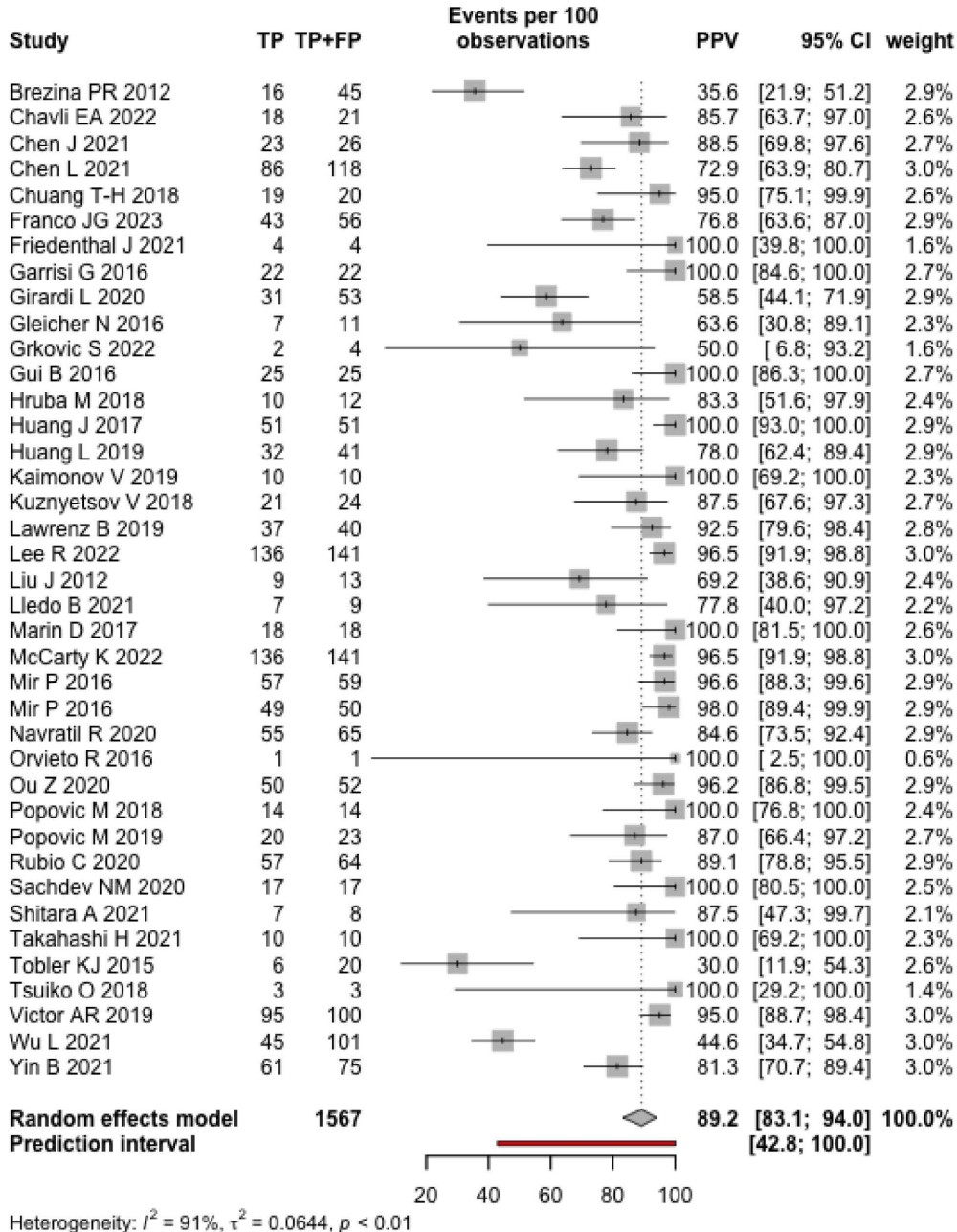

| Study | TP | TP+FP | Events per 100 observations | PPV | 95% CI | weight |
|---|---|---|---|---|---|---|
| Brezina PR 2012 | 16 | 45 | | 35.6 | [21.9; 51.2] | 2.9% |
| Chavli EA 2022 | 18 | 21 | | 85.7 | [63.7; 97.0] | 2.6% |
| Chen J 2021 | 23 | 26 | | 88.5 | [69.8; 97.6] | 2.7% |
| Chen L 2021 | 86 | 118 | | 72.9 | [63.9; 80.7] | 3.0% |
| Chuang T-H 2018 | 19 | 20 | | 95.0 | [75.1; 99.9] | 2.6% |
| Franco JG 2023 | 43 | 56 | | 76.8 | [63.6; 87.0] | 2.9% |
| Friedenthal J 2021 | 4 | 4 | | 100.0 | [39.8; 100.0] | 1.6% |
| Garrisi G 2016 | 22 | 22 | | 100.0 | [84.6; 100.0] | 2.7% |
| Girardi L 2020 | 31 | 53 | | 58.5 | [44.1; 71.9] | 2.9% |
| Gleicher N 2016 | 7 | 11 | | 63.6 | [30.8; 89.1] | 2.3% |
| Grkovic S 2022 | 2 | 4 | | 50.0 | [6.8; 93.2] | 1.6% |
| Gui B 2016 | 25 | 25 | | 100.0 | [86.3; 100.0] | 2.7% |
| Hruba M 2018 | 10 | 12 | | 83.3 | [51.6; 97.9] | 2.4% |
| Huang J 2017 | 51 | 51 | | 100.0 | [93.0; 100.0] | 2.9% |
| Huang L 2019 | 32 | 41 | | 78.0 | [62.4; 89.4] | 2.9% |
| Kaimonov V 2019 | 10 | 10 | | 100.0 | [69.2; 100.0] | 2.3% |
| Kuznyetsov V 2018 | 21 | 24 | | 87.5 | [67.6; 97.3] | 2.7% |
| Lawrenz B 2019 | 37 | 40 | | 92.5 | [79.6; 98.4] | 2.8% |
| Lee R 2022 | 136 | 141 | | 96.5 | [91.9; 98.8] | 3.0% |
| Liu J 2012 | 9 | 13 | | 69.2 | [38.6; 90.9] | 2.4% |
| Lledo B 2021 | 7 | 9 | | 77.8 | [40.0; 97.2] | 2.2% |
| Marin D 2017 | 18 | 18 | | 100.0 | [81.5; 100.0] | 2.6% |
| McCarty K 2022 | 136 | 141 | | 96.5 | [91.9; 98.8] | 3.0% |
| Mir P 2016 | 57 | 59 | | 96.6 | [88.3; 99.6] | 2.9% |
| Mir P 2016 | 49 | 50 | | 98.0 | [89.4; 99.9] | 2.9% |
| Navratil R 2020 | 55 | 65 | | 84.6 | [73.5; 92.4] | 2.9% |
| Orvieto R 2016 | 1 | 1 | | 100.0 | [2.5; 100.0] | 0.6% |
| Ou Z 2020 | 50 | 52 | | 96.2 | [86.8; 99.5] | 2.9% |
| Popovic M 2018 | 14 | 14 | | 100.0 | [76.8; 100.0] | 2.4% |
| Popovic M 2019 | 20 | 23 | | 87.0 | [66.4; 97.2] | 2.7% |
| Rubio C 2020 | 57 | 64 | | 89.1 | [78.8; 95.5] | 2.9% |
| Sachdev NM 2020 | 17 | 17 | | 100.0 | [80.5; 100.0] | 2.5% |
| Shitara A 2021 | 7 | 8 | | 87.5 | [47.3; 99.7] | 2.1% |
| Takahashi H 2021 | 10 | 10 | | 100.0 | [69.2; 100.0] | 2.3% |
| Tobler KJ 2015 | 6 | 20 | | 30.0 | [11.9; 54.3] | 2.6% |
| Tsuiko O 2018 | 3 | 3 | | 100.0 | [29.2; 100.0] | 1.4% |
| Victor AR 2019 | 95 | 100 | | 95.0 | [88.7; 98.4] | 3.0% |
| Wu L 2021 | 45 | 101 | | 44.6 | [34.7; 54.8] | 3.0% |
| Yin B 2021 | 61 | 75 | | 81.3 | [70.7; 89.4] | 3.0% |
| **Random effects model** | | **1567** | | **89.2** | **[83.1; 94.0]** | **100.0%** |
| **Prediction interval** | | | | | **[42.8; 100.0]** | |

Heterogeneity: $I^2 = 91\%$, $\tau^2 = 0.0644$, $p < 0.01$

**Fig 2.** Continued.

Lin evaluated outcomes based on level of mosaicism (Low: 21–49%; High: 50–80% abnormal cells) and found higher miscarriage rates in the high-level group compared to the low-level group (31% vs 5%) but similar live birth rates (45% vs 36%) [57]. All 46 patients with ongoing pregnancies had amniocentesis confirming euploid karyotypes. Similarly, Rubino et al (2018) transferred low mosaic embryos in 33 patients resulting in a 51.5% euploid live birth rate compared to a 48.5% live birth rate among euploid screened embryos [94].

c. Positive predictive value of mosaic embryos

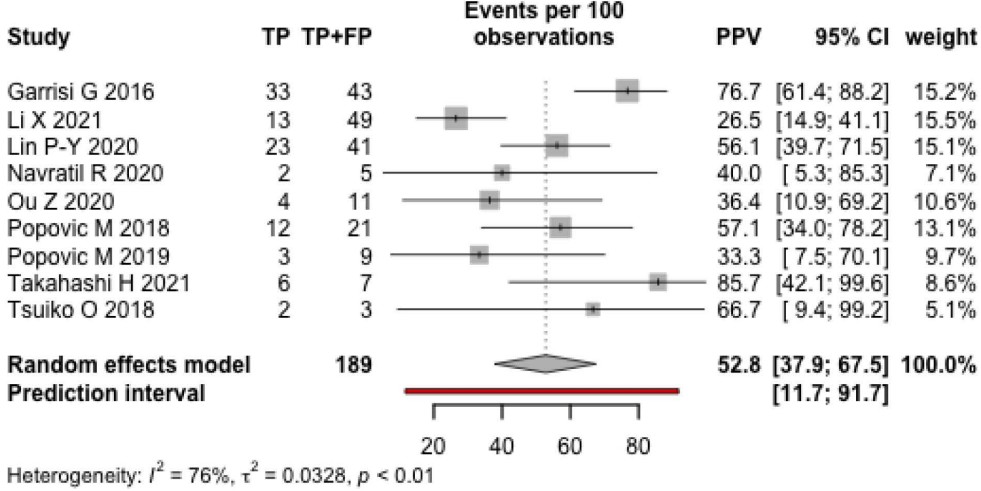

**Fig 2.** Continued.

## Quality evaluation

The quality of the studies was largely regarded as low risk of bias for the index test (S11 Table). However, the risk of bias for the interpretation of the reference standard was unclear or high risk due to lack of blinding, and incomplete or missing description of the genetic platform for evaluating the POC, pre- or post-natal genetic tests. As many included patients did not receive confirmatory testing, the risk of bias was high for patient flow and timing resulting from a loss to follow-up.

## Discussion

In this systematic review and meta-analysis, we demonstrate that among studies that report pregnancy outcomes after a euploid ET, the risk of a false negative is very low. While there was no significant difference in measures of accuracy by genetic platform used for PGT-A analysis, unsurprisingly, blastocyst biopsy had higher predictive value than cleavage-stage embryo, which is the standard of testing.

An ideal study of diagnostic accuracy would first perform the index test and compare results for all embryos. In the case of euploid ET, this is impossible as many will either fail to implant or result in an early pregnancy loss where it is not feasible to collect tissue samples for cytogenetic analysis, leading to an incalculable denominator [124,125]. More-over, most patients do not transfer all their euploid embryos, and many decline prenatal and postnatal genetic testing. Despite being encouraged to undergo confirmation testing, the study conducted by Tiegs (2021) revealed that only 10% of patients with ongoing pregnancies underwent either CVS or amniocentesis [103]. As many patients decline invasive prenatal testing, universal neonatal or cord blood sampling may be a more feasible option to confirm the initial screening result and allow for further investigation into causes of misdiagnosis (specifically mosaicism and technical factors like contamination).

Notably, there are several studies that did not meet the initial inclusion criteria based on the lack of genetic evaluation of POC, be it from a miscarriage, amniocentesis, CVS, or neonatal examination. For inclusion "healthy birth" was insufficient to describe the diagnostic accuracy of PGT-A. In the STAR trial, where women were randomized to PGT-A or untested blastocyst transfer, karyotyping was not performed on miscarriages [126]. The multicentre RCT conducted by Yan (2021) did not perform any genetic evaluation on products of conception and there was no report of pre- or postnatal testing [127]. As a result, the accuracy of euploid transfer is difficult to ascertain.

**Table 2. Mosaicism level and karyotype concordance of whole embryo/ICM studies.**

| Study | Mosaicism level | Karyotype concordance | Notes |
|---|---|---|---|
| Brezina PR 2012[38] | Not described | | |
| Chavli EA 2022[39] | 20-80% | 8 partial concordance and 1 complete concordance | Detected mosaicism in 59% of embryos |
| Chen J 2021[40] | 40-70%; 30% for chr13, chr16, chr18, chr 21 | 3 false were mosaic embryos in the TE 30–50%; 2 embryos had partial karyotype concordance with ICM | |
| Chen L 2021[41] | Not described | 5209/5267 (97.6%) of chromosome sets were consistent by ploidy | 9 embryos failed to amplify |
| Chuang T-H 2018[42] | 20-80% | 25/29 embryos were completely concordant | Segmental imbalances > 10 Mb was considered segmental aneuploidy; 2 failed amplifications in TE1 and 2 failed in ICM. 30% mosaic in TE and 40% mosaic in ICM |
| Franco JG 2023[43] | Not described | | |
| Friedenthal J 2021[44] | Not described | | Completed single cell DNA sequencing; 320/433 cells were successfully sequenced. Mosaicism was identified in 31.4% of all euploid embryos |
| Garrisi G 2016[45] | 10-90% | | Of aneuploid embryos 0/22 abnormal. Of mosaic embryos 10/43 were euploid ICM but not broken down by complexity and severity of mosaicism |
| Girardi L 2020[46] | Not described | | Segmental aneuploidies > 10 Mb |
| Gleicher N 2016[7] | Not described | 99.9% per chromosome and 98.7% for the full karyotype | |
| Grkovic S 2022[47] | Low-level: 20–40%; high-level > 40 to < 80% | | Segmental aneuploidies > 10 Mb |
| Gui B 2016[48] | Not described | 86.5% consistency; 1 euploid TE was complex mosaic for 3 involved chromosomes (30%, 50%, 60%). There were 3 embryos that were partially inconsistent with ICM due to mosaicism | |
| Hruba M 2018[49] | Not described | 67% consistency; 7 samples were aneuploid, 1 mosaic (70%). Two samples were more complex aneuploid arrangements and there were 2 euploid ICM | |
| Huang J 2017[50] | Not described | 50 blastocysts had complete concordance in all segments biopsied. 1 embryo had partial concordance between initial biopsy and ICM | |
| Huang L 2019[51] | 30-60% | 2 embryos were aneuploid but had inconsistent karyotype abnormalities between TE and ICM; 6 embryos with partial concordance between embryo and TE. | Minimum resolution > 10 Mb |
| Kaimonov V 2019[52] | Not described | 3 aneuploid embryos were incompletely consistent by karyotype | |
| Kuznyetsov V 2018[53] | Not described | 2 aneuploid embryos were incompletely consistent by karyotype | |
| Lawrenz B 2019[54] | Not described | 6 embryos had partially discordant karyotypes between TE and ICM | 3 embryos had no DNA detected |
| Lee R 2022[55] | Not described | 20/21 euploid, 85/87 aneuploid and 51/54 segmental aneuploid were concordant. Remaining blastocysts were mosaic | |
| Li X 2021[56] | Normal (possible mosaic): 20–50%; Abnormal (possible mosaic): 50–80% | | All included embryos were mosaic on initial biopsy ranging from 27–68%. False positives only occurred with mosaic embryos < 52%. If set threshold to 50%, then only 1 false positive. If set threshold to 40%, only 5 false positive |

*(Continued)*

**Table 2.** (Continued)

| Study | Mosaicism level | Karyotype concordance | Notes |
|---|---|---|---|
| Lin P-Y 2020[57] | Low-level: >20% to <50%; high-level ≥50 to ≤80% | | Of 27 high-level mosaic embryos, 11 were euploid, 10 mosaic and 6 aneuploid. Of 14 low-level mosaic embryos, 7 were mosaic, 7 euploid and none aneuploid. |
| Liu J 2012[58] | Not described | 8 embryos had complete karyotype concordance. 1 embryo had a partial concordance | |
| Lledo B 2021[59] | Not described | 7/9 embryos had complete karyotype concordance | |
| Marin D 2017[24] | Not described | | 2 embryos had non-concurrent results |
| McCarty K 2022[60] | Not set. Gain if deviation more than 50 | | Chromosomal deletions and duplications ≥5 Mb. 54 blastocysts with deletions, 21 euploid embryos and 87 aneuploid embryos validated |
| Mir P 2016[61] | Not described | | Segmental imbalances were excluded (n=6). Abnormal was considered when the Log2 ratio was increased above 0.3 threshold; 12/50 day-3 embryos were mosaic aneuploid and 1/50 day 3 embryos was euploid. 9/59 blastocysts were mosaic aneuploid and 2/59 were euploid |
| Navratil R 2020[62] | 30-80% | 25/65 aneuploid embryos were partially concordant; 3/65 were completely discordant in abnormality with initial aneuploid result | Resolution of 4 Mb; 31 segmental errors ranging from 5–150 Mb. The euploid false negative was mosaic in the embryo 40–50%. |
| Orvieto R 2016[6] | Not described | | 2 embryos were inconclusive |
| Ou Z 2020[63] | 20-80% | 7/54 had partially discordant aneuploid karyotypes | True positive aneuploid blastocysts had aneuploid ranging 40–70%. False positive mosaic ranged from 30–50%. |
| Popovic M 2018[25] | | 5/12 mosaic embryos had complete concordant aneuploid karyotype; 12/14 aneuploid embryos had complete concordant karyotype | 1 euploid embryo was mosaic for multiple chromosomes. 5 embryos were non-informative for their biopsy or ICM |
| Popovic M 2019[64] | 3/10 cells | 16/21 aneuploid embryos were had complete concordance in karyotype, while 2/3 mosaic embryos had complete concordance. 5/21 aneuploid embryos had partial concordance and 1 mosaic embryo was completely discordant. | Resolution of >10 Mb. |
| Rubio C 2020[65] | 30-70% | 87.5% of embryos were consistent for ploidy | Resolution of >10 Mb. 64/81 embryos were informative for TE, ICM and cfDNA. 1 embryo was discordant in sex of embryo |
| Sachdev NM 2020[66] | Low-level: >20% to <40%; high-level ≥40% to <80% | | 16 samples were uninterpretable due to chaotic results. 2 euploid embryos were mosaic in ICM (20–50%) |
| Shitara A 2021[67] | 20-80% | 1/7 aneuploid embryos was completely concordant; 4/7 were partially concordant | 2 aneuploid embryos were incompletely consistent by karyotype; |
| Takahashi H 2021[68] | 1-99% | | 7 mosaic embryos on TE biopsy ranged from 10–80%. Mosaicism ≤20% was not detected in BE. 9 aneuploid embryos were completely concordant. 1 completely discordant and 3 partially concordant |
| Tobler KJ 2015[69] | Not described | 5/6 aneuploid embryos were completely concordant; 1 embryo was completely discordant | |
| Tsuiko O 2018[70] | 20-80% | 2/2 aneuploid embryos were completely concordant. 2/3 mosaic with high level mosaicism ≥50% was detected in ICM | 1 mosaic embryo with multiple involved chromosomes (20–50%) were not detected in ICM. One embryo with mosaicism 50% was detected at 20% in ICM. A third embryo with mosaicism 80% was detected at 70% in ICM. |

*(Continued)*

**Table 2.** (Continued)

| Study | Mosaicism level | Karyotype concordance | Notes |
|---|---|---|---|
| Victor AR 2019[71] | 20-80% | 79/93 aneuploid embryos were completely concordant. 14 were partially concordant. | Resolution of 20 Mb |
| Wu L 2021[72] | 30-70% | | |
| Yin B 2021[73] | 30-70% for trisomy 13, 16, 18, 21; 40–60% for other chromosomes | 42/75 embryos were completely concordant and 34/59 had partial concordance. | |

In this meta-analysis, among studies that reported a genetic evaluation of pregnancy tissue, the misdiagnosis rate was 0.2% for euploid embryos, but higher for mosaic (21.7%) and aneuploid embryos (10.6%). Importantly, the prediction interval for mosaic transfers and aneuploid transfers are extremely wide. These wide intervals indicate a lack of certainty in the true misdiagnosis rate for mosaic and aneuploid embryos and suggest the need for future high-quality studies to inform practice. Two studies knowingly transferred aneuploid embryos tested at multiple centres with limited sample sizes; [7,75] the original study by Gleicher's group transferred embryos that performed PGT-A with various genetic platforms (with limited ability to detect mosaicism) [7]. Three studies were only unblinded to the PGT result after the ET [97,98,103]. These differences may explain the heterogeneity detected. Among studies that performed either POC testing or amniocentesis, many did not describe the genetic platform of evaluation. Importantly, of the 15 studies that tested POC to confirm the karyotype of the pregnancy loss [4,5,75,85,87,88,97,100,102,105,108,110,113,114,122], 11 described the method of genetic analysis. The studies by Maxwell (2016), Wang (2018) and Werner (2014) used a combination of SNP and cell culture with G Banding [5,108,113], which is no longer recommended as the gold standard for cytogenetic analysis of POC due to the risk of amplification failure and maternal contamination [128].

Among mosaic ET, the misdiagnosis rate was 21.7%. Similarly, for whole embryo studies, the PPV was 52.8%, indicating that embryos classified as mosaic are in fact euploid in at least 20% of cases but even as high as over 50%. This wide range is likely attributed to the criterion for consideration of mosaicism and the distinction between low and high levels. Several studies are demonstrating encouraging pregnancy outcomes after mosaic ET [129], leading to controversy on the definition of aneuploid/euploid embryos and calling into question clinical practice decisions on how to handle these embryos. The incidence of mosaicism ranges from 5–15%, which varies by clinic, embryology practices and testing facilities [130]. This additional layer of complexity in the use of PGT-A as a selection tool means that genetic counselling is essential to guide patients in decision-making: 1. To discard potentially healthy embryos and proceed with another IVF cycle due to lack of remaining embryos, or 2. To transfer with the potential for either failed implantation or pregnancy loss with the associated financial and emotional burden, as well as the delayed time to successful pregnancy. Certainly, it is important to closely monitor outcomes of these pregnancies to determine a more reliable estimate of diagnostic accuracy [130,131]. In fact, in a survey of IVF centres, 95% (151/159) recommended prenatal diagnostic testing for confirmation and follow up [132]. For the purposes of our study, we considered a mosaic or aneuploid result to be concordant, even if the chromosomes involved were non-concordant (i.e., concordance by ploidy not by individual chromosome). Several studies also performed mixture studies with cell lines of varying fractions of euploid cells to validate mosaic embryos to determine the threshold of the platform to detect mosaicism [19,20,23,25,27,30,31]. In trophectoderm biopsies, several chromosomes may be involved in the aneuploidy and to some degree of mosaicism, which has been investigated in five studies [40,45,56,57,72]. It is of utmost importance for labs to perform these studies prior to initiating clinical testing, which will guide in the determination of a "safe" threshold for embryo transfer. The decision to re-biopsy the embryo is challenging, as there are unclear benefits nor specific indications to perform one. Among the studies that had multiple TE biopsies and could compare to the ICM/WE for both aneuploid or mosaic results in TE1 [6,24,25,46,50,56,59,62,71], there was

**Table 3. Characteristics for pregnancy outcomes studies.**

| Study ID | Publication type | Country | Number of patients | Number of embryos tested | Number of transferrable embryos | Patient population | Mean female age (years) | Study design | Stage of embryo development at biopsy | Initial diagnosis | Index test: Method of aneuploidy detection (initial TE biopsy) | Reference standard: Method of aneuploidy detection | Reference standard |
|---|---|---|---|---|---|---|---|---|---|---|---|---|---|
| Aharon D 2022[74] | Conference abstract | USA | 28 | Not described | Not described | Patients who underwent IVF with PGT-A with subsequent mosaic embryo transfer | Not described | Case series | Blastocyst | Mosaic | NGS | Microarray (POC), not described for amniocentesis | Amniocentesis or POC |
| Barad DH 2022[75] | Full text | USA | 69 | Not described | Not described | Patients who underwent IVF with PGT-A, for whom only abnormal embryos were available for transfer | 41.4 (±3.98) | Prospective cohort study | Blastocyst (Day 5–7) | Aneuploid or mosaic | NGS for all but 3 cycles (array CGH or SNP microarray) | Not described | Prenatal diagnosis (amniocentesis/ CVS), POC (microarray) or postnatal microarray (not described which was performed among the live births) |
| Besser AG 2019[76] | Full text | USA | 98 | Not described | Not described | Patients who underwent IVF with PGT-A, for whom only mosaic embryos were available | Not described | Retrospective case series | Blastocyst | Mosaic | NGS | Not described | Amniocentesis |
| Chamayou S 2015[77] | Conference abstract | Italy | 7 | 39 | 10 | Patients who underwent combined PGT-M and PGT-A for β-hemoglobinopathies | Not described | Case series | Blastocyst (Day 5–6) | Unaffected euploid | NGS | Not described | Prenatal diagnosis (not described) |
| Chen D 2020[78] | Full text | China | 12 | 112 | 37 | Patients who underwent combined PGT-M and PGT-A for α- and β-thalassemia | Range: 26–36 | Case series | Blastocyst (Day 5–6) | Unaffected euploid | NGS | Not described | Amniocentesis |

*(Continued)*

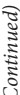

| Study ID | Publication type | Country | Number of patients | Number of embryos tested | Number of transferrable embryos | Patient population | Mean female age (years) | Study design | Stage of embryo development at biopsy | Initial diagnosis | Index test: Method of aneuploidy detection (initial TE biopsy) | Reference standard: Method of aneuploidy detection | Reference standard |
|---|---|---|---|---|---|---|---|---|---|---|---|---|---|
| Daina G 2015[16] | Full text | Spain | 7 | 62 | 13 | Patients who underwent combined PGT-M and PGT-A for cystic fibrosis (n=4), polycystic kidney disease (n=3), arrhythmogenic right ventricular dysplasia/cardiomyopathy (n=1), sickle cell disease (n=1) | 33.6 | Case series | Cleavage stage (Day 3) | Unaffected euploid | Metaphase comparative genomic hybridization (mCGH) | Not described | Postnatal genetic diagnosis |
| Fernandez Sanguino A[79] | Conference abstract | Spain | 2022 | 532 | Not described | Patients who underwent IVF with PGT-A, for whom only mosaic embryos were available | Not described | Retrospective case series | Blastocyst | Mosaic | NGS | Not described | Amniocentesis |
| Friedenthal J 2020[4] | Full text | USA | 1997 | Not described | Not described | Not described | 35.8 | Retrospective case series | Blastocyst | Euploid | Array CGH (n=846); NGS (n=1151) | Not described | Karyotype of POC/amniocentesis/CVS or neonatal exam |
| Gao Y 2022[80] | Full text | China | 7 | Not described | Not described | Not described | Not described | Case control | Blastocyst | Mosaic | NGS | Single cell multiomics sequencing | Postnatal genetic diagnosis |
| Gleicher N 2016[7] | Full text | USA | 8 | Not described | Not described | Not described | Not described | Prospective case series | Blastocyst | Aneuploid | NGS | Array CGH | Amniocentesis or CVS |
| Hu X 2024[81] | Full text | China | 20 | 89 | 47 | Patients who underwent combined PGT-M and PGT-A for small copy number variants | Range 21–36 | Prospective case series | Blastocyst (Day 5–6) | Unaffected euploid | NGS | Karyotype (G-banding) | Amniocentesis |

(Continued)

| Study ID | Publication type | Country | Number of patients | Number of embryos tested | Number of transferrable embryos | Patient population | Mean female age (years) | Study design | Stage of embryo development at biopsy | Initial diagnosis | Index test: Method of aneuploidy detection (initial TE biopsy) | Reference standard: Method of aneuploidy detection | Reference standard |
|---|---|---|---|---|---|---|---|---|---|---|---|---|---|
| Huang C 2022[82] | Full text | China | 3 | 10 | 4 | Patients who underwent IVF with either combined PGT-M for neurofibromatosis (n=1) or PGT-SR for a Robertsonian translocation (n=2) | 30.7 | Prospective case series | Blastocyst (Day 5–6) | Unaffected euploid | NGS | Not described | Amniocentesis |
| Huang J 2015[83] | Full text | China | 6 | 58 | 7 | Patients who underwent PGT-A±PGT-SR for recurrent pregnancy loss or parental karyotype abnormality (n=5) | Not described | Retrospective case series | Cleavage stage (Day 3) followed by blastocyst | Euploid balanced | Array CGH or SNP microarray | Karyotype | Amniocentesis |
| Katz-Jaffe M 2023[84] | Abstract | USA | Not described | Not described | Not described | Patients who underwent single euploid embryo transfer | 35.8±3.8 | Retrospective case series | Blastocyst | Euploid | NGS | Karyotype and/or SNP microarray | POC |
| Kim JG 2021[85] | Conference abstract | USA | Not described | Not described | Not described | Not described | Not described | Retrospective case series | Not described | Euploid | NGS | Not described | POC |
| Klimczak AM 2020[86] | Full text | USA | 1139 | Not described | Not described | Patients who conceived after single euploid embryo transfer and underwent non-invasive prenatal testing | 35.3 (±4) for normal NIPT; 37.1±2 for abnormal NIPT; Range 18–50 | Retrospective cohort | Not described | Euploid | Not described | Not described | Amniocentesis or CVS and neonatal exam |
| Lin P-Y 2020[57] | Full text | Taiwan | 108 | Not described | Not described | Patients who underwent IVF with PGT-A for infertility, recurrent pregnancy loss, or combined PGT-M | Not described | Retrospective case series | Blastocyst (Day 5–6) | Mosaic | NGS | Karyotype | Amniocentesis |

(Continued)

**Table 3.** (Continued)

| Study ID | Publication type | Country | Number of patients | Number of embryos tested | Number of transferrable embryos | Patient population | Mean female age (years) | Study design | Initial diagnosis | Stage of embryo development at biopsy | Index test: Method of aneuploidy detection (initial TE biopsy) | Reference standard: Method of aneuploidy detection | Reference standard |
|---|---|---|---|---|---|---|---|---|---|---|---|---|---|
| Luo KL 2015[87] | Conference abstract | China | 101 | Not described | Not described | Patients with unexplained recurrent pregnancy loss, abnormal CGH on products of conception testing, or advanced maternal age | Not described | Retrospective case series | Euploid | Blastocyst (Day 5–6) | SNP microarray | CGH | POC |
| Ma GC 2016[88] | Full text | Taiwan | 21 | 144 | 74 | Patients who underwent their first IVF cycle for infertility | 36.0; Range: 29–42 | Prospective cohort study | Euploid | Blastocyst (Day 5) | Array CGH | Array CGH | POC |
| Ma X 2021[89] | Full text | China | 258 | 1189 | 538 | Patients who underwent IVF with PGT-A±PGT-SR for infertility, recurrent pregnancy loss, previous karyotypically abnormal conception (n=116, control group), or parental karyotype abnormality (n=142, study group) | 31.0±5.78 for experimental group; 32.4±4.68 for control group | Prospective cohort study | Euploid balanced | Blastocyst (Day 5–6) | NGS | Karyotype | Amniocentesis |
| Maxwell SM 2016[5] | Full text | USA | 76 | Not described | Not described | Patients who underwent IVF with PGT-A for infertility with subsequent pregnancy loss after euploid embryo transfer | 35.5±5.5 | Retrospective case control | Euploid | Blastocyst (Day 5–6) | Array CGH | SNP x 10 pregnancies; Cell culture, G banding x 10 pregnancies | POC |

*(Continued)*

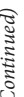

**Table 3.** (Continued)

| Study ID | Publication type | Country | Number of patients | Number of embryos tested | Number of transferrable embryos | Patient population | Mean female age (years) | Study design | Stage of embryo development at biopsy | Initial diagnosis | Index test: Method of aneuploidy detection (initial TE biopsy) | Reference standard: Method of aneuploidy detection | Reference standard |
|---|---|---|---|---|---|---|---|---|---|---|---|---|---|
| Morales Sabater R 2023[90] | Conference abstract | Spain | Not described | Not described | Not described | Children born after SET for PGT-A | Not described | Retrospective case control | Blastocyst | Euploid or mosaic | NGS | Not described | Amniocentesis, or postnatal karyotype, and neonatal exam |
| Mykytenko D 2018[91] | Conference abstract | Ukraine | 20 | Not described | Not described | Not described | Not described | Retrospective case series | Blastocyst | Euploid | Array CGH | NGS | POC |
| Ou Z 2022[92] | Full text | China | 23 | 143 | 65 | Patients who underwent IVF with combined PGT-A± PGT-SR for advanced reproductive age or parental karyotype abnormality (balanced or Robertsonian translocation, or chromosome inversion) | 31.1±4.1 | Retrospective case series | Blastocyst (Day 5–6) | Euploid balanced | NGS and SNP microarray | Not described | Prenatal diagnosis or POC |
| Pozzoni M 2022[93] | Conference abstract | Italy | 39 | Not described | Not described | Patients who underwent IVF with PGT-A, for whom only abnormal embryos were available for transfer with resulting pregnancy | Not desfraffrcribed | Retrospective case control | Blastocyst | Mosaic | NGS | Not described | CVS or amniocentesis |
| Rubino P 2018[94] | Conference abstract | USA | 99 | Not described | Not described | Patients who underwent IVF with PGT-A, for whom no euploid embryos were available for transfer | 39.1±6.4 | Retrospective case control | Blastocyst | Mosaic | NGS | Not described | CVS or amniocentesis |

*(Continued)*

**Table 3.** (Continued)

| Study ID | Publication type | Country | Number of patients | Number of embryos tested | Number of transferrable embryos | Patient population | Mean female age (years) | Study design | Stage of embryo development at biopsy | Initial diagnosis | Index test: Method of aneuploidy detection (initial TE biopsy) | Reference standard: Method of aneuploidy detection | Reference standard |
|---|---|---|---|---|---|---|---|---|---|---|---|---|---|
| Ruttanajit T 2016[95] | Full text | China, Thailand | 7 | 49 | 29 | Patients who underwent IVF with PGT-A for infertility, advanced reproductive age, or balanced translocation | Not described | Case series | Blastocyst (Day 5–6) | Euploid | Array CGH | Karyotype | Not described |
| Satirapod C 2019[96] | Full text | Thailand | 15 | 106 | Not described | Patients who underwent combined PGT-M and PGT-A for β-thalassemia/ hemoglobin E disease | 34.8±3.56 | Retrospective case series | Blastocyst | Unaffected euploid | Array CGH (n=5) or NGS (n=10) | Not described | Amniocentesis and postnatal cord blood sampling |
| Scott RT 2012[97] | Full text | USA | 146 | 255 | Not applicable | Patients who underwent IVF with PGT-A for infertility | 34.0±4.4 | Prospective non-selection study | Cleavage stage (Day 3; n=113); Blastocyst (Day 5; n=142) | Blinded results (non-selection) | SNP microarray | SNP microarray | Postnatal genetic diagnosis |
| Shen X 2019[98] | Conference abstract | China | Not described | Not described | 103 | Patients who underwent PGT-M for single gene defect; PGT-A completed after delivery | Not described | Retrospective case series | Blastocyst | Unaffected (PGT-A results determined after delivery) | NGS | Not described | Prenatal diagnosis or karyotype detection of the newborn |
| Spinella F 2018[27] | Full text | Italy | 327 | Not described | Not described | Patients who underwent IVF with PGT-A for infertility, advanced reproductive age or translocation, for which no euploid embryos were available for transfer | 37.6; Range: 39–47 | Prospective cohort study | Blastocyst (Day 5–6) | Mosaic | Array CGH and NGS | Not described | Amniocentesis and/or chorionic villi sampling |

(Continued)

| Study ID | Publication type | Country | Number of patients | Number of embryos tested | Number of transferrable embryos | Patient population | Mean female age (years) | Study design | Stage of embryo development at biopsy | Initial diagnosis | Index test: Method of aneuploidy detection (initial TE biopsy) | Reference standard: Method of aneuploidy detection | Reference standard |
|---|---|---|---|---|---|---|---|---|---|---|---|---|---|
| Spinella F 2023[99] | Conference abstract | International | Not described | Not described | Not described | Patients who underwent IVF with PGT-A with mosaic ET | Not described | Retrospective case series | Blastocyst (Day 5–7) | Mosaic | NGS | Not described | Prenatal testing and postnatal exam |
| Tan Y 2014[100] | Full text | China | 395 (NGS = 128; SNP = 177) | 1512 | 666 | Patients who underwent IVF with PGT-A for advanced reproductive age, recurrent pregnancy loss or parental karyotype abnormality | 32.1; Range: 20–44 | Retrospective case series | Blastocyst (Day 5–6) | Euploid balanced | SNP microarray (n = 1058); NGS (n = 454) | Karyotype for PGT | Amniocentesis or peripheral blood samples for babies |
| Tao X 2020[101] | Conference abstract | USA | Not described | Not described | Not applicable | Not described | Not described | Nested case series from non-selection study | Not specified | Blinded results (non-selection) | NGS | NGS | CVS, amniocentesis or newborn buccal swabs |
| Tiegs AW 2016[102] | Full text | USA | 520 | Not described | Not described | Not described | 35.9 | Prospective case series | Blastocyst (Day 5–7) | Euploid | Array CGH | Not described | POC from pregnancy loss tissue (array CGH), amniocentesis, neonatal karyotype |
| Tiegs AW 2021[103] | Full text | USA | 402 | 2110 | Not applicable | Patients who underwent their first IVF cycle for infertility | 34.9±4.0 | Prospective non-selection study | Blastocyst | Blinded results (non-selection) | NGS | Microarray | Neonatal exam, neonatal karyotype, amniocentesis, products of conception |
| Treff NR 2011[104] | Full text | USA | 15 | 122 | 39 | Patients who underwent combined PGT-SR with PGT-A for parental karyotype abnormality | Not described | Prospective case series | Blastocyst | Euploid balanced | SNP microarray | SNP microarray | Neonatal karyotype (buccal DNA) |

(Continued)

| Study ID | Publication type | Country | Number of patients | Number of embryos tested | Number of transferrable embryos | Patient population | Mean female age (years) | Study design | Stage of embryo development at biopsy | Initial diagnosis | Index test: Method of aneuploidy detection (initial TE biopsy) | Reference standard: Method of aneuploidy detection | Reference standard |
|---|---|---|---|---|---|---|---|---|---|---|---|---|---|
| Vesela K 2019[105] | Conference abstract | Czech Republic | 59 | Not described | Not described | Patients who experienced pregnancy loss and underwent dilation and curettage | Not described | Retrospective case series | Blastocyst (Day 5–6) | Euploid | NGS | Array CGH | Products of conception |
| Victor AR 2019[106] | Full text | UK | Not described | Not described | Not described | Patients who underwent IVF with PGT-A, for whom no euploid embryos were available for transfer | Not described | Prospective case series | Blastocyst | Mosaic | NGS | Not described | Amniocentesis, physical exam (1 twin gestation leading to demise of both babies secondary to PPROM) |
| Volozonoka L 2018[107] | Full text | Latvia | 9 | 62 | 20 | Patients who underwent IVF with combined PGT-M and PGT-A for single gene defect | 34.4±2.8 | Prospective case series | Blastocyst (Day 5) | Unaffected euploid | Array CGH | Not described | Postnatal genetic diagnosis |
| Wang J 2018[108] | Full text | China | 11 | 107 | 23 | Patients who underwent IVF with combined PGT-SR and PGT-A for parental karyotype abnormality (Robertsonian translocation) | 30.6±2.62 | Case series | Blastocyst (Day 5–6) | Euploid balanced | SNP microarray | Conventional G-banding karyotype | Amniocentesis and POC |
| Wang J 2023[109] | Full text | China | 25 | Not described | Not described | Patients who underwent combined PGT-M and PGT-A for α-thalassemia | 30.1±3.30; Range 23–39 | Case series | Blastocyst | Unaffected euploid | SNP microarray | Not described | Amniocentesis |

(Continued)

**Table 3.** (Continued)

| Study ID | Publication type | Country | Number of patients | Number of embryos tested | Number of transferrable embryos | Patient population | Mean female age (years) | Study design | Stage of embryo development at biopsy | Initial diagnosis | Index test: Method of aneuploidy detection (initial TE biopsy) | Reference standard: Method of aneuploidy detection | Reference standard |
|---|---|---|---|---|---|---|---|---|---|---|---|---|---|
| Wang Y 2021[110] | Full text | China | 9 | 34 | 17 | Patients who underwent IVF with combined PGT-M and PGT-A for de novo autosomal dominant kidney disease | Range: 23–34 | Prospective case series | Blastocyst (Day 5–6) | Unaffected euploid | NGS | SNP microarray for POC; Not described for amniocentesis | Amniocentesis and POC |
| Wang Y 2023[111] | Full text | China | 8 | 45 | 18 | Patients who underwent IVF with combined PGT-M and PGT-A for Charcot-Marie-Tooth disease | Range 26–38 | Prospective case series | Blastocyst (Day 5–6) | Unaffected euploid | NGS | Not described | Amniocentesis |
| Wells D 2009[112] | Conference abstract | UK, USA | 97 | 432 | 194 | Not described | 38.3 | Case series | Blastocyst | Euploid | CGH | Not described | Not described |
| Werner MD 2014[113] | Full text | USA | Not described | Not described | Not described | Not described | Not described | Retrospective case series | Blastocyst | Euploid | qPCR | G banding conventional karyotype and SNP microarray for 4 POC | POC |
| Wiltshire AM 2021[114] | Conference abstract | USA | Not described | Not described | Not described | Patients who underwent IVF with PGT-A and subsequently experienced pregnancy loss | Not described | Retrospective case series | Not described | Euploid | Not described | SNP microarray | POC |
| Yang J 2021[115] | Full text | China | 10 | 23 | 19 | Patients who underwent combined PGT-M and PGT-A with previous history of at least two molar pregnancies | Range: 27–34 | Prospective case series | Blastocyst (Day 5–6) | Biparental disomy euploid | NGS | SNP microarray | Amniocentesis and neonatal karyotype |

*(Continued)*

Table 3. (Continued)

| Study ID | Publication type | Country | Number of patients | Number of embryos tested | Number of transferrable embryos | Patient population | Mean female age (years) | Study design | Stage of embryo development at biopsy | Initial diagnosis | Index test: Method of aneuploidy detection (initial TE biopsy) | Reference standard: Method of aneuploidy detection | Reference standard |
|---|---|---|---|---|---|---|---|---|---|---|---|---|---|
| Yao Z 2023[116] | Full text | China | 17 | Not described | Not described | Patients who underwent PGT-A±PGT-SR for recurrent pregnancy loss, recurrent implantation failure, or parental karyotype abnormality who had a live birth | 30.5±4.82 | Retrospective case series | Blastocyst (Day 5–6) | Euploid or mosaic (n = 1) | NGS | NGS | Neonatal karyotype (whole blood) |
| Zhai F 2022[117] | Full text | China | 109 | 540 | 233 | Patients who underwent combined PGT-SR and PGT-A for parental karyotype abnormality with history of either pregnancy loss or infertility | 30.3±3.42 | Retrospective case series | Blastocyst | Euploid balanced | NGS and SNP microarray for CNV for carrier status | Karyotype | Amniocentesis |
| Zhang L 2019[118] | Full text | China | 348 | Not described | Not described | Patients who underwent IVF with PGT-A for infertility, parental karyotype abnormality, unexplained recurrent pregnancy loss, previous karyotypically abnormal conception, or advanced reproductive age | 31.4±4.2 (mosaic on reanalysis); 31.3±4.6 (euploid on reanalysis) | Retrospective case series | Blastocyst (Day 5–6) | Euploid | Array CGH | Not described | Amniocentesis and neonatal exam |

*(Continued)*

| Study ID | Publi-cation type | Country | Number of patients | Num-ber of embryos tested | Number of trans-ferrable embryos | Patient population | Mean female age (years) | Study design | Stage of embryo develop-ment at biopsy | Initial diagno-sis | Index test: Method of aneu-ploidy detection (initial TE biopsy) | Reference standard: Method of aneuploidy detection | Reference standard |
|---|---|---|---|---|---|---|---|---|---|---|---|---|---|
| Zhang S 2017[119] | Full text | China | 11 | 68 | 26 | Patients who underwent com-bined PGT-SR and PGT-A for parental karyo-type abnormality with history of either recurrent pregnancy loss, infertility or pre-vious karyotyp-ically abnormal conception | Range: 25–36 | Prospec-tive case series | Blastocyst (Day 5–6) | Euploid balanced | SNP microarray | Karyotype | Amniocentesis |
| Zhang S 2019[120] | Full text | China | 4 | 18 | 8 | Patients who underwent com-bined PGT-SR and PGT-A for parental chromo-somal inversion with history of infertility, recur-rent pregnancy loss, or karyotyp-ically abnormal conception | Not described | Prospec-tive case series | Blastocyst (Day 5–6) | Euploid balanced | SNP microarray | Karyotype | Amniocentesis or postnatal cord blood |
| Zhang S 2021[121] | Full text | China | 12 | 59 | 22 | Patients who underwent combined PGT-M, PGT-SR and PGT-A for couples where both part-ners were carriers for monogenic disease, one of whom carried a parental karyotype abnormality. They had history of either infertility or previous karyo-typically abnormal pregnancy | Not described | Case series | Blastocyst (Day 5–6) | Unaf-fected euploid balanced | SNP microarray | Karyotype and Sanger sequencing | Amniocentesis or postnatal cord blood |

*(Continued)*

| Study ID | Publication type | Country | Number of patients | Number of embryos tested | Number of transferrable embryos | Patient population | Mean female age (years) | Study design | Stage of embryo development at biopsy | Initial diagnosis | Index test: Method of aneuploidy detection (initial TE biopsy) | Reference standard: Method of aneuploidy detection | Reference standard |
|---|---|---|---|---|---|---|---|---|---|---|---|---|---|
| Zhang YX 2020[122] | Full text | China, Malaysia, Thailand | Not described | Not described | Not described | Patients who underwent IVF with PGT-A for infertility, recurrent pregnancy loss, previous karyotypically abnormal conception, or combined PGT-M or PGT-SR | 31.8±6.4 (mosaic); 34.5±5.8 (euploid) | Prospective cohort study | Blastocyst (Day 5–6) | Euploid or mosaic | NGS (2 centres) and array CGH (1 centre) | Karyotyping for POC (not described), microarray for amniocentesis | Amniocentesis or POC |
| Zhou Z 2018[123] | Full text | China | 12 | 118 | 19 | Patients who underwent combined PGT-SR and PGT-A for parental karyotype abnormalities | Range: 26–37 | Case series | Cleavage stage (Day 3) | Euploid | NGS and array CGH | Karyotype | Amniocentesis and neonatal karyotype (buccal cells) |

CGH: Comparative genomic hybridization; CVS: Chorionic villus sampling; NGS: Next generation sequencing; PCR: Polymerase chain reaction; PGT-A: Preimplantation genetic testing for aneuploidy; PGT-M: Preimplantation genetic testing for monogenic diseases; PGT-SR: Preimplantation genetic testing for structural rearrangements; POC: Products of conception; SNP: Single nucleotide polymorphism

significant variability in TE2 or TE3 and how it reflected the "true" ICM/WE result. We would, therefore, not recommend a re-biopsy for this indication, as supported by the European Society of Human Reproduction and Embryology (ESHRE) [132]. Rather, patients should be counselled of the possibility of the risk of inaccuracies in PGT-A testing [130].

While the absolute false negative rate of PGT-A cannot be known in clinical practice, as presumed euploid embryos that are transferred but fail to implant cannot be retested, the false negative rate can be estimated by performing genetic testing on products of conception from clinical pregnancy losses, as well as genetic testing of ongoing pregnancies with suspected aneuploidy syndromes. Several groups have reported discordance rates, with failure to detect mosaic embryos, monosomy and polyploidy ranging from 0.1% to 23% [4,102,113], depending on whether the pregnancy resulted in a spontaneous abortion or live birth. There is biologic plausibility that euploid-screened embryos are not implanting because the test is wrong, however, failed implantation cannot be solely attributed to the ploidy status, and uterine factors, other embryonic factors must also be taken into consideration. One study noted that the clinical error rate was significantly higher after pregnancy loss compared to the pregnancies that resulted in a live birth (13–23% vs 0.1–0.4%) [4], which is

a. Euploid embryo transfer

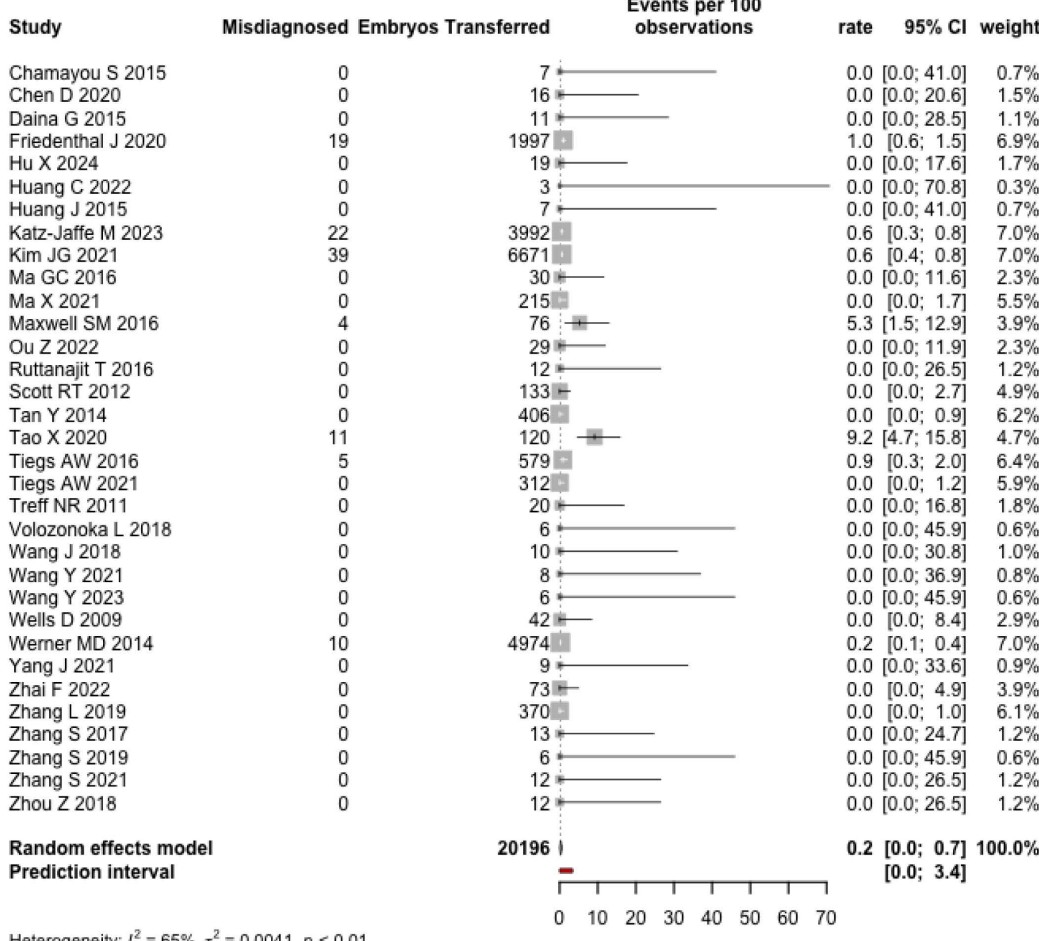

**Fig 3. Forest plots for pregnancy outcomes: misdiagnosis rate. a. Euploid embryo transfer. b. Aneuploid embryo transfer. c. Non-selection embryo transfer. d. Mosaic embryo transfer.**

b. Aneuploid embryo transfer

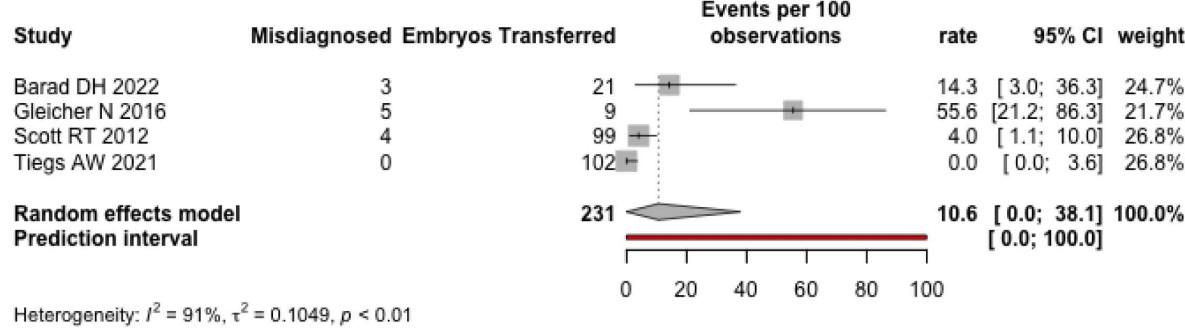

c. Non-selection embryo transfer

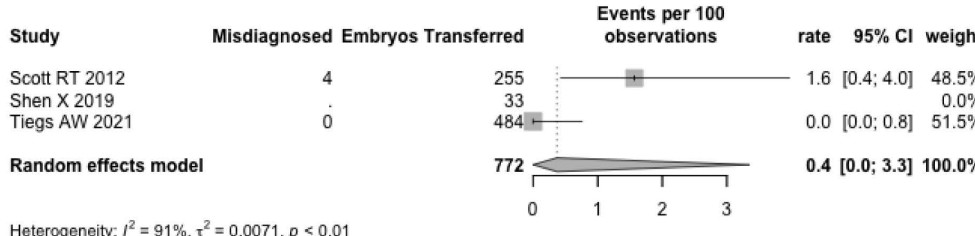

d. Mosaic embryo transfer

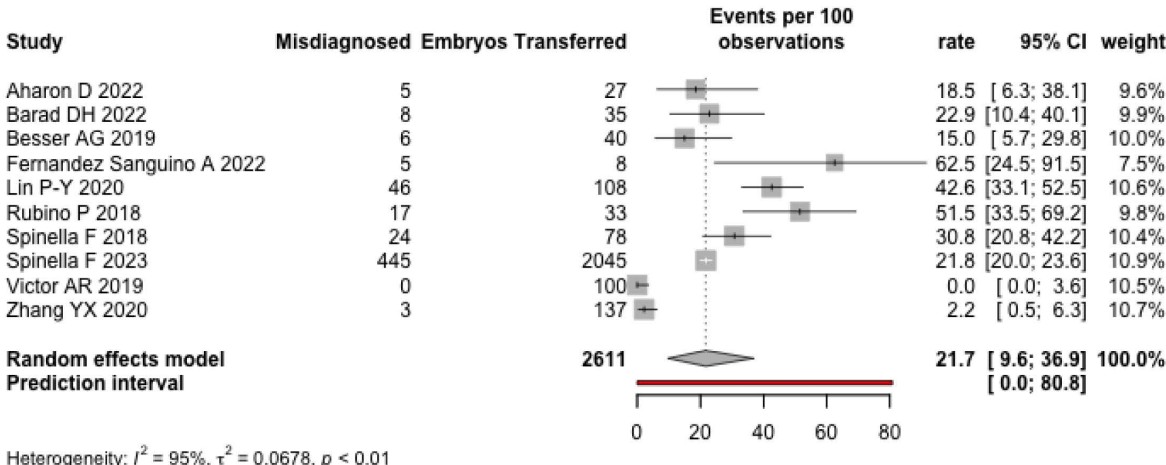

**Fig 3.** Continued.

understandable as most chromosomal aneuploidies are incompatible with life [133]. Encouraging patients to collect their POC tissue after a spontaneous pregnancy loss at home, sending the POC for cytogenetic analysis after D&C, or working with PGT-A reference labs to offer POC testing at no cost for adequate follow-up would help elucidate the misdiagnosis rate (or reason for pregnancy loss of an aneuploid conception) and potentially reduce the risk of detection bias. This would also serve to avoid unnecessary investigations and interventions for failed "euploid embryo transfer" in the case of

**Table 4. Mosaicism level and karyotype concordance of pregnancy outcome studies.**

| Study | Mosaicism level | Karyotype concordance | Notes |
|---|---|---|---|
| Aharon D 2022[74] | Not described | | |
| Barad DH 2022[75] | Not described | 7 whole chromosome aneuploid mosaic were non-concordant; 2 segmental aneuploid mosaic was non-concordant. 2 whole chromosome aneuploid and 1 segmental aneuploid were euploid upon testing pregnancy | |
| Besser AG 2019[76] | 20-80% | | Prevalence of mosaicism was 28.4%; 5 cycles were double embryo transfer |
| Chamayou S 2015[77] | Not described | | |
| Chen D 2020[78] | Not described | | |
| Daina G 2015[16] | Not described | | |
| Fernandez Sanguino A[79] | Not described | | 39 low risk mosaic embryos of which 8 were transferred. |
| Friedenthal J 2020[4] | Not described | 11 cases of discrepant diagnoses with aCGH: 2 were below the threshold of detection, 3 mosaic, and one contamination; 8 cases of discrepant diagnoses with NGS in POC with 2 WCA, 3 mosaic, and 2 segmental aneuploid. | |
| Gao Y 2022[80] | Not described | | Resolution > 10 Mb |
| Gleicher N 2016[7] | Not described | | |
| Hu X 2024[81] | Not described | | Resolution 57 Kb using SNP |
| Huang C 2022[82] | Not described | | |
| Huang J 2015[83] | Not described | | |
| Katz-Jaffe M 2023[84] | Not described | | 12 cases confirmed fetal mosaicism; triploidy detected in 4 cases, WCA in 6 cases |
| Kim JG 2021[85] | Not described | 4 cases undetected by PGT-A; 21 cases with deletions/duplications 5.02 Mb-111kB; 10 cases with whole chromosome mosaic abnormalities; 4 cases of tetraploidy that elude detection by NGS. | |
| Klimczak AM 2020[86] | Not described | | 1 case of Turner mosaicism (80%) |
| Lin P-Y 2020[57] | 20-80%; Low level < 50%; High level > 50% | | 83 low-level mosaic; 25 high-level mosaic transfers; 37 LB in low-mosaic; 9 LB in high-mosaic |
| Luo KL 2015[87] | Not described | | |
| Ma GC 2016[88] | Not described | | 9 double embryo transfer cycles |
| Ma X 2021[89] | 30-70%; Low level < 50%; High level ≥50% | | 20.6% of embryos exhibited mosaicism. |
| Maxwell SM 2016[5] | 20-80% | Misdiagnoses: 47,XX, + 7; mosaic trisomy 21; mosaic trisomy 13; mosaic trisomy 11 | |
| Morales Sabater R 2023[90] | 25-50% | | 61.4% of embryos were mosaic in the range of 25–39%; 38.6% of embryos were mosaic in the range of 40–50% cells tested |
| Mykytenko D 2018[91] | <50% | 2 cases of mosaicism in POC after euploid transfer (T13, M20) | |
| Ou Z 2022[92] | >30% | | Resolution > 4 Mb |
| Pozzoni M 2022[93] | IQR 30–40% | | |
| Rubino P 2018[94] | 20-80%; Low level < 50%; High level > 50% | | |

*(Continued)*

 

**Table 4.** (Continued)

| Study | Mosaicism level | Karyotype concordance | Notes |
|---|---|---|---|
| Ruttanajit T 2016[95] | 20-70% | | 13% of embryos exhibited mosaicism |
| Satirapod C 2019[96] | Unable to detect | | |
| Scott RT 2012[97] | Not described | | 55 healthy live births and 3 losses from euploid cohort; 4 live births and 2 losses from the aneuploid cohort. |
| Shen X 2019[98] | Not described | Misdiagnoses: T22 (n = 1); segmental chromosomal aneuploidy (4.06 Mb-191 Mb, n = 6); mosaic whole chromosome aneuploidy (20–41%, n = 28); mosaic segmental chromosomal aneuploidy (n = 8); combined segmental aneuploidy with mosaic aneuploidy (n = 16) | Resolution > 4 Mb; 30.4% of embryos were mosaic whole chromosomal aneuploidy; 6.5% of embryos were segmental chromosomal aneuploid. |
| Spinella F 2018[27] | Array CGH: log2 ratio between 3x SD 0.08 +/- 0.04 and 0.033 +/- 0.02; NGS: copy number value between 2 and 3 or 2 and 1 | | All healthy live births occurred with mosaicism levels 30–50%. 54 embryos either did not implant or led to early losses for mosaicism level between 30–60%. |
| Spinella F 2023[99] | 20-80% | | Resolution > 5 Mb |
| Tan Y 2014[100] | Not described | | Resolution > 1 Mb |
| Tao X 2020[101] | Not described | 3 POC samples had maternal contamination; Misdiagnoses: 3 non-concurrent results; | |
| Tiegs AW 2016[102] | Unable to detect | All but one live birth was apparently euploid and of the correct gender; Misdiagnoses: 1 contamination from embryologist, 4 speculated to be secondary to mosaicism or technical error | |
| Tiegs AW 2021[103] | Not described | Microdeletion 1.1 Mb on chromosome 13 (below threshold of detection for PGT-A) detected on one POC; 3.5% of embryos were whole chromosomal mosaic and 8.8% had a segmental abnormality | |
| Treff NR 2011[104] | Not described | | |
| Vesela K 2019[105] | Not described | | |
| Victor AR 2019[106] | 20-80% | 2 microdeletions below level of detection (84.11 Kb; 1 copy < 100 kb) 1 balanced translocation | Resolution > 20 Mb (occasionally < 2 Mb) |
| Volozonoka L 2018[107] | Not described | | |
| Wang J 2018[108] | Not described | | |
| Wang J 2023[109] | Not described | | |
| Wang Y 2021[110] | >30% | | Resolution > 10 Mb |
| Wang Y 2023[111] | >30% | | Resolution > 10 Mb |
| Wells D 2009[112] | Not described | | |
| Werner MD 2014[113] | Not described | | 1 tetraploid, 2 monosomic, 7 trisomic gestations; 4 cases of mosaicism |
| Wiltshire AM 2021[114] | Not described | | 3 trisomies, 2 partial duplications, 2 mosaic trisomies, 1 triploid |
| Yang J 2021[115] | Not described | | |
| Yao Z 2023[116] | 20-80% | | Resolution > 10 Mb |
| Zhai F 2022[117] | Not described | | 15.6% of embryos were mosaic |
| Zhang L 2019[118] | >20% | | |
| Zhang S 2017[119] | Not described | | |

*(Continued)*

**Table 4.** (Continued)

| Study | Mosaicism level | Karyotype concordance | Notes |
|---|---|---|---|
| Zhang S 2019[120] | Not described | | |
| Zhang S 2021[121] | Not described | | |
| Zhang YX 2020[122] | 20-80% | | Resolution > 10 Mb |
| Zhou Z 2018[123] | Not described | | |

a misdiagnosis. Thus, while the reported misdiagnosis rate per ET is 0.2%, indicating that approximately 2/1000 presumed euploid embryos are actually aneuploid and may explain a pregnancy loss or failed implantation, this is can only be a rough estimate and may only be "tip of the iceberg" as misdiagnosis of failed implantation cases simply cannot be quantified.

There were two studies that attempted to identify sources of misdiagnosis [4,102]. Friedenthal et al. (2020) described potential sources of discrepancy for PGT-A results, specifically biologic sources (likely attributed to mosaicism) and test error [4]. They reported on DNA fingerprinting to confirm embryologist contamination, which was only performed in one case where there was sex discordance between PGT-A result and live birth. In a more recent conference presentation at ESHRE a contamination rate of 0.44% using SNPs to detect non-embryonic DNA was found in nearly 50,000 analyzed biopsies, though was as high as 7.7% in one clinic [134]. This study did not identify the source of contamination (e.g., from technician or parental origins). Finally, Dong et al evaluated the risk of contamination in embryos fertilized by conventional IVF and found maternal contamination in 0.83% from granulosa cells and 0% for sperm [135]. While rare, it would be prudent to conduct a study investigating the incidence of technician or parental contamination.

Gleicher et al (2016) suggested that the risk of a false positive test with PGT-A may be as high as 55% [7]. While not every embryo is resampled, and most embryos labelled aneuploid are not transferred, a misdiagnosis rate this high would have considerable implications, including discarding potentially healthy embryos and reduction in cumulative live birth rates. In a subsequent study by the same group the misdiagnosis rate was lower (14.3%) [75], with very small sample sizes. However, when considering the transfer of aneuploid-screened embryos in the two well-designed non-selection studies [97,103], this misdiagnosis rate declines further to 6%, which is more reassuring, and likely more realistic. From ICM or WE studies, the positive predictive value of an aneuploid-screened embryo validated against their ICM or WE was 84%, meaning that up to 16% of embryos are in fact euploid, and would otherwise be discarded [6,24,25,46,50,56,59,62,71].

Two studies biopsied embryos at the cleavage stage and compared the results to ICM or WE after extended culture [38,69]. If the embryo failed to develop to blastocyst stage, they were excluded from analysis, thereby introducing possible selection bias. Aneuploid embryos may be more likely to arrest in development and were therefore more likely be excluded compared to euploid embryos [3,38,136]. Brezina (2012) demonstrated that 60% of embryos initially classified as aneuploid failed to develop to blastocyst, compared to the 40% blastulation rate among euploid embryos [38]. Similarly, Popovic (2019) investigated blastocyst outgrowths and found that euploid embryos were significantly more likely to continue developing in extended culture compared to aneuploid embryos [64]. This observed attrition may therefore result in a lower prevalence of aneuploidy compared to euploidy in these studies, which would lead to a reduction in the positive predictive value.

## Limitations

A significant challenge of this meta-analysis was the limited number of randomized controlled trials and non-selection studies available for inclusion, which would be the highest calibre of studies to answer our research objectives. The diagnostic accuracy from studies evaluating pregnancy outcomes were largely immeasurable or only partially verified (evaluating

exclusively aneuploid or exclusively euploid embryos). Partial verification bias of diagnostic tests occurs when a proportion of the embryos are compared to the reference standard [137], as is the case in studies evaluating euploid-screened embryos where few pregnancies are compared to a gold standard (amniocentesis/CVS/neonatal karyotype), or when only aneuploid-classified embryos are dissected and compared to the ICM/whole embryo. This effect biases euploid results to an increased sensitivity and lower specificity [138,139]. Moreover, the positive predictive value is particularly impacted by a high prevalence of outcome (aneuploidy), which would be set by the authors if exclusively evaluating one outcome. In application to the WE/ICM studies included, most embryos evaluated were donated "aneuploid" or "poor quality" embryos. Due to the limited number of donated euploid-screened embryos, the results presented may be an inflation of the actual PPV, and calls into question the number of potentially healthy embryos we are discarding as a result of an erroneous "euploid" result.

There is also a concern about the generalizability of these results as many of the included studies are published from the same clinics. A recently published study investigated the variation in euploidy rate and live birth rates based on the genetics labs [140]. This study evaluated four high-volume genetics companies and found that the lab with the highest euploidy rate also had the highest live birth rate. The authors suggested that this may largely be due to quality control, including processing and data analysis. These results may also be confounded by high-volume centres sending a large proportion of their samples to a single centre with superior embryology and biopsy techniques. The reassuring measures of diagnostic accuracy found in this study may reflect programs with the highest competence in embryo biopsy and the PGT-A pipeline. It is therefore prudent for each embryology and genetics lab to continue to perform quality control measures with validation pre-clinically, as well as follow-up post embryo transfer.

## Conclusion

The overall accuracy of PGT-A is excellent, and patients can be counselled that the results are reliable for euploid and aneuploid classifications. The risk of false negative result leading to an aneuploid conception appears to be very low. When we conducted sensitivity analyses to evaluate the impact of various PGT-A genetic platforms on diagnostic accuracy, particularly for aneuploid-screened embryos, there was no difference detected. However, the accuracy for mosaic embryos is much lower, with high possibility of healthy pregnancy, with consideration to either re-biopsy or transfer with adequate counselling. Clinicians should be aware that the estimates of diagnostic accuracy are biased by missing data from failed implantation, pregnancy losses and limited pre- or postnatal diagnostic testing.

## Supporting information

**S1 Fig. Forest plots for cell line studies.**
(DOCX)

**S2 Fig. Forest plots for cell lines studies subgroup analysis: NGS vs other genetic platform.**
(DOCX)

**S3 Fig. Forest plots for cell lines studies subgroup analysis: Conference abstract vs full text.**
(DOCX)

**S4 Fig. Forest plots for whole embryo or ICM studies: Measures of diagnostic accuracy.**
(DOCX)

**S5 Fig. Forest plots for whole embryo or ICM studies subgroup analysis: NGS vs other genetic platform.**
(DOCX)

**S6 Fig. Forest plots for whole embryo or ICM studies subgroup analysis: Conference abstract vs full text.**
(DOCX)

**S7 Fig. Forest plots for whole embryo or ICM studies subgroup analysis: Whole embryo vs ICM.**
(DOCX)

**S8 Fig. Forest plots for whole embryo or ICM studies subgroup analysis: Blastocyst vs cleavage stage embryos.**
(DOCX)

**S1–11 Tables.** Characteristics for cell line studies, two-by-two tables all study types, and quality assessments for all study types.
(DOCX)

**S1 File. Medline search strategy.**
(DOC)

**S2 File. Excluded studies with reasons for exclusion.**
(DOCX)

**S3 File. Cell line supplement results and discussion.**
(DOCX)

**S4 File. Prospero registration.**
(PDF)

**S5 File. PRISMA DTA Checklist.**
(DOC)

**S6 File. Included studies.**
(XLSX)

**S7 File. Full citation list.**
(ZIP)

## Acknowledgments

We would like to thank John Matelski for his assistance with statistical planning.

## Author contributions

**Conceptualization:** Vanessa Bacal, Heather Shapiro, Rhonda Zwingerman, Crystal Chan.

**Data curation:** Vanessa Bacal, Angela Li, Urvi Rana, Eleni Philipoppolous.

**Formal analysis:** Vanessa Bacal, Lisa Avery.

**Funding acquisition:** Vanessa Bacal.

**Investigation:** Vanessa Bacal, Angela Li, Heather Shapiro.

**Methodology:** Vanessa Bacal, Rhonda Zwingerman, Lisa Avery, Alina Palermo, Eleni Philipoppolous, Crystal Chan.

**Project administration:** Vanessa Bacal.

**Resources:** Eleni Philipoppolous.

**Supervision:** Heather Shapiro, Crystal Chan.

**Validation:** Lisa Avery, Alina Palermo.

**Visualization:** Lisa Avery.

**Writing – original draft:** Vanessa Bacal.

**Writing – review & editing:** Vanessa Bacal, Angela Li, Heather Shapiro, Urvi Rana, Rhonda Zwingerman, Lisa Avery, Alina Palermo, Crystal Chan.

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
