## [Decision Letter · Decision Letter 0]

7 Jan 2025

PONE-D-24-53437A systematic review and meta-analysis of the diagnostic accuracy after preimplantation genetic testing for aneuploidyPLOS ONE

Dear Dr. Bacal,

Thank you for submitting your manuscript to PLOS ONE. After careful consideration, we feel that it has merit but does not fully meet PLOS ONE’s publication criteria as it currently stands. Therefore, we invite you to submit a revised version of the manuscript that addresses the points raised during the review process.

We look forward to receiving your revised manuscript.

Kind regards,

Qinghua Shi

Academic Editor

PLOS ONE

**Journal Requirements:**

Our project was supported by funding from the Department of Obstetrics and Gynaecology at Sinai Health, Toronto, Ontario, Canada.

4. As required by our policy on Data Availability, please ensure your manuscript or supplementary information includes the following: 

Reviewers' comments:

Reviewer's Responses to Questions

**Comments to the Author**

1. Is the manuscript technically sound, and do the data support the conclusions?

Reviewer #1: Partly

2. Has the statistical analysis been performed appropriately and rigorously? 

Reviewer #1: Yes

3. Have the authors made all data underlying the findings in their manuscript fully available?

Reviewer #1: Yes

4. Is the manuscript presented in an intelligible fashion and written in standard English?

Reviewer #1: Yes

5. Review Comments to the Author

**Reviewer #1:**  Below are my detailed comments and suggestions for the authors:

1. Expand on the controversies surrounding the clinical application of PGT-A, particularly the challenges and implications of mosaic embryo diagnoses.

2. Provide additional background on the clinical consequences of false-positive and false-negative results for patient counseling.

3. Clarify how the inclusion and exclusion criteria were applied to studies with mixed designs or incomplete data reporting.

4. Address how the authors handled studies with unclear risk of bias as assessed by the QUADAS-2 tool and its potential impact on the conclusions.

5. Justify the inclusion of conference abstracts and discuss how this might affect the robustness of the analysis.

6. Acknowledge and discuss the substantial heterogeneity in some analyses, particularly in mosaic embryo predictive values (e.g., high I² values).

7. Provide additional context on the clinical significance of predictive intervals and discuss how they influence decision-making in clinical practice.

8. Highlight any notable differences in diagnostic accuracy between genetic platforms (e.g., NGS vs. aCGH).

9. Expand on the clinical implications of the findings, particularly the management of mosaic embryos, and explore alternative strategies such as complementary diagnostic methods or second biopsies.

10. Discuss the potential influence of publication bias, particularly the possible underrepresentation of studies with negative findings.

11. Elaborate on how study selection criteria (e.g., inclusion of donated embryos) might introduce partial verification bias or limit generalizability.

12. Ensure technical terms, such as predictive intervals, are briefly explained for a broader audience, including non-genetics specialists.

13. Review the manuscript for potential redundancy and enhance the clarity of key findings to improve readability.

14. Consider adding annotations to figures and tables to highlight clinically relevant thresholds or differences between embryo categories.

15. Provide a brief explanation of how subgroup analyses (e.g., biopsy stage, genetic platforms) contribute to the overall conclusions.

6. PLOS authors have the option to publish the peer review history of their article (what does this mean? ). If published, this will include your full peer review and any attached files.

**Do you want your identity to be public for this peer review?** For information about this choice, including consent withdrawal, please see our Privacy Policy .

Reviewer #1: No

---

## [Author Response · Author response to Decision Letter 1]

13 Feb 2025

Response to the editors

Thank you for considering our manuscript for publication in Plos One and taking the time for such a careful review. We have addressed the reviewer’s comments below and made the requested edits. We look forward to hearing a positive response.

Reviewer #1: Below are my detailed comments and suggestions for the authors:

1. Expand on the controversies surrounding the clinical application of PGT-A, particularly the challenges and implications of mosaic embryo diagnoses.

Response: We have addressed some of the controversies, particularly with mosaicism in the discussion.

Line 344: The incidence of mosaicism ranges from 5-15%, which varies by clinic, embryology practices and testing facilities.(130) This additional layer of complexity in the use of PGT-A as a selection tool means that genetic counselling is essential to guide patients in decision-making: 1. To discard potentially healthy embryos and proceed with another IVF cycle due to lack of remaining embryos, or 2. To transfer with the potential for either failed implantation or pregnancy loss with the associated financial and emotional burden, as well as the delayed time to successful pregnancy. Certainly, it is important to closely monitor outcomes of these pregnancies to determine a more reliable estimate of diagnostic accuracy (130,131). In fact, in a survey of IVF centres, 95% (151/159) recommended prenatal diagnostic testing for confirmation and follow up.(132)

Line 363: Among the studies that had multiple TE biopsies and could compare to the ICM/WE for both aneuploid or mosaic results in TE1,(6,24,25,46,50,56,59,62,71) there was significant variability in TE2 or TE3 and how it reflected the “true” ICM/WE result. We would, therefore, not recommend a re-biopsy for this indication, as supported by the European Society of Human Reproduction and Embryology (ESHRE).(132) Rather, patients should be counselled of the possibility of the risk of inaccuracies in PGT-A testing.(130)

2. Provide additional background on the clinical consequences of false-positive and false-negative results for patient counseling.

Response: We added to the background:

Line 87: Those that screen positive (i.e., aneuploid or chromosomally abnormally) are not selected for transfer and often discarded

We added to the discussion paragraph:

Line 381: Encouraging patients to collect their POC tissue after a spontaneous pregnancy loss at home, sending the POC for cytogenetic analysis after D&C, or working with PGT-A reference labs to offer POC testing at no cost for adequate follow-up would help elucidate the misdiagnosis rate (or reason for pregnancy loss of an aneuploid conception) and potentially reduce the risk of detection bias. This would also serve to avoid unnecessary investigations and interventions for failed “euploid embryo transfer” in the case of a misdiagnosis. Thus, while the reported misdiagnosis rate per ET is 0.2%, indicating that approximately 2/1000 presumed euploid embryos are actually aneuploid and may explain a pregnancy loss or failed implantation, this is can only be a rough estimate and may only be “tip of the iceberg” as misdiagnosis of failed implantation cases simply cannot be quantified.

3. Clarify how the inclusion and exclusion criteria were applied to studies with mixed designs or incomplete data reporting.

Response: We included all studies that described their methodology to sufficient detail in their validation for replication. In the context of mixed designed (i.e. cell line study pre-validation and whole embryo study or pregnancy outcomes), all data were extracted separately and meta-analyzed in the appropriate group.

Line 119: We included all studies with sufficient detail in their validation for replication. We also included abstracts if full length manuscripts were not available, provided there was sufficient information for a two-by-two table

Line 189: Studies that conducted a mixed design (i.e. cell line pre-clinical validation and pregnancy outcomes) were extracted and meta-analyzed separately in their appropriate category.

4. Address how the authors handled studies with unclear risk of bias as assessed by the QUADAS-2 tool and its potential impact on the conclusions.

Response: We elected not to perform separate subgroup analyses based on QUADAS-2 tool as we anticipated unclear or high risk of bias for all of pregnancy outcomes studies just by their inherent design (apart from the non-selection studies, which were analyzed separately as planned). We anticipated high bias for the abstracts due to missing data, which were also analyzed as a subgroup analysis.

5. Justify the inclusion of conference abstracts and discuss how this might affect the robustness of the analysis.

Response: To mitigate the risk of publication bias and to be more inclusive, we decided to pre-emptively include abstracts to address this, provided there was enough information to extract for both the narrative review and/or meta-analysis.

Line 120: As we anticipated that many studies, particularly the pre-clinical designs, would demonstrate high validity and would not proceed to publication, we elected to include abstracts if full length manuscripts were not available, provided there was sufficient information for a narrative review and/or a two-by-two table.

6. Acknowledge and discuss the substantial heterogeneity in some analyses, particularly in mosaic embryo predictive values (e.g., high I² values).

Response:

Yes, the heterogeneity of PPV of aneuploid and mosaic embryos is high, as is the embryo transfer for aneuploid, non-selection or mosaic embryos. To help readers interpret this heterogeneity we have added information to the analysis section,

Line 168: We have reported I2, a measure of heterogeneity across studies where values > 75% indicate high variability across study results. To contextualise the heterogeneity, we have also computed prediction intervals, which indicates the range of effect sizes we would expect to see in a new study. Wide prediction intervals indicate high uncertainty in future results.

7. Provide additional context on the clinical significance of predictive intervals and discuss how they influence decision-making in clinical practice.

Additional context has been provided in the discussion,

Line 324: These wide intervals indicate a lack of certainty in the true misdiagnosis rate for mosaic and aneuploid embryos and suggest the need for future high quality studies to inform practice.

Discussion of clinical practice also provided on lines 344-351

8. Highlight any notable differences in diagnostic accuracy between genetic platforms (e.g., NGS vs. aCGH).

Response: Thank you for your comment. In the results section, we previously mentioned:

Line 219: We performed sensitivity analyses investigating the impact of PGT-A platforms (NGS vs other platforms), reference comparators of ICM biopsy or WE, and publication of results (conference abstract versus full text). There were no differences comparing NGS to other platforms or full-text publications compared to conference abstracts (S5, S6 Fig). Sample sizes in the other platforms groups and conference proceedings were too small to determine their impact. Measures of diagnostic accuracy were slightly higher when comparing ICM biopsy to the WE with less heterogeneity (S7 Fig). While the impact of stage of biopsy revealed a higher overall accuracy at blastocyst compared to cleavage embryos (84.3 vs 60.0), the heterogeneity was still high (> 80%) (S8 Fig).

We added to the discussion:

Line 298: While there was no significant difference in measures of accuracy by genetic platform used for PGT-A analysis, unsurprisingly, blastocyst biopsy had higher predictive value than cleavage-stage embryo, which is the standard of testing.

9. Expand on the clinical implications of the findings, particularly the management of mosaic embryos, and explore alternative strategies such as complementary diagnostic methods or second biopsies.

Response: See response to point 1 above

10. Discuss the potential influence of publication bias, particularly the possible underrepresentation of studies with negative findings.

Response: Publication bias is an unavoidable issue in all systematic reviews and meta-analyses. Analogous to how the misdiagnosis rate of euploid embryos is difficult to ascertain because failed implantation cases cannot be studied, the publication bias problem is not quantifiable. We have tried to be overly inclusive with abstracts and not just published manuscripts to partially address this issue (see response to question 5). Additionally, it is difficult to ascertain, in the context of this study question, what would actually constitute a “negative finding”. PGT-A has been utilized and evolving since the 1990s and widely applied in clinical medicine, even before randomized controlled trials could demonstrate its clinical benefit. The question then begs, do researchers “want” to publish studies that confirm it as a valid test or to disprove it as an invalid/inaccurate test. The included studies have ranged from nearly 100% accuracy to nearly 50% accuracy (as is the case with Gleicher’s study). However, we assumed that the preclinical testing that confirmed its accuracy (including the cell-lines and whole embryo/ICM studies) would get presented but not published due to the lack of “excitement” in their content, which is why we elected to include published abstracts.

11. Elaborate on how study selection criteria (e.g., inclusion of donated embryos) might introduce partial verification bias or limit generalizability.

Response: We have added information in the limitations section:

Line 437: In application to the WE/ICM studies included, most embryos evaluated were donated “aneuploid” or “poor quality” embryos. Due to the limited number of donated euploid-screened embryos, the results presented may be an inflation of the actual PPV, and calls into question the number of potentially healthy embryos we are discarding as a result of an erroneous “euploid” result.

12. Ensure technical terms, such as predictive intervals, are briefly explained for a broader audience, including non-genetics specialists.

Response: We have added additional text in the analysis section, please see the response to item 6.

13. Review the manuscript for potential redundancy and enhance the clarity of key findings to improve readability.

Response: We have made edits throughout the manuscript to minimize redundancy as suggested and highlighted key findings.

14. Consider adding annotations to figures and tables to highlight clinically relevant thresholds or differences between embryo categories.

Response: Thank you very much to this reviewer for all the excellent suggestions which we have addressed in the manuscript. We do not understand what the reviewer is asking for here, as we believe our figures and annotations to be quite clear for scientific and clinical interpretation. If there is something specific that should be edited in production review, we are happy with any suggestions.

15. Provide a brief explanation of how subgroup analyses (e.g., biopsy stage, genetic platforms) contribute to the overall conclusions.

Response: Thank you for your comment. See above for comment #8

---

## [Decision Letter · Decision Letter 1]

12 Mar 2025

A systematic review and meta-analysis of the diagnostic accuracy after preimplantation genetic testing for aneuploidy

PONE-D-24-53437R1

Dear Dr. Vanessa Bacal,

We’re pleased to inform you that your manuscript has been judged scientifically suitable for publication and will be formally accepted for publication once it meets all outstanding technical requirements.

Kind regards,

Qinghua Shi

Academic Editor

PLOS ONE

Additional Editor Comments (optional):

Reviewers' comments:

Reviewer's Responses to Questions

**Comments to the Author**

1. If the authors have adequately addressed your comments raised in a previous round of review and you feel that this manuscript is now acceptable for publication, you may indicate that here to bypass the “Comments to the Author” section, enter your conflict of interest statement in the “Confidential to Editor” section, and submit your "Accept" recommendation.

Reviewer #1: All comments have been addressed

2. Is the manuscript technically sound, and do the data support the conclusions?

Reviewer #1: Yes

3. Has the statistical analysis been performed appropriately and rigorously? 

Reviewer #1: Yes

4. Have the authors made all data underlying the findings in their manuscript fully available?

Reviewer #1: Yes

5. Is the manuscript presented in an intelligible fashion and written in standard English?

Reviewer #1: Yes

6. Review Comments to the Author

Reviewer #1: (No Response)

7. PLOS authors have the option to publish the peer review history of their article (what does this mean? ). If published, this will include your full peer review and any attached files.

**Do you want your identity to be public for this peer review?** For information about this choice, including consent withdrawal, please see our Privacy Policy .

Reviewer #1: **Yes: ** Bo Xu

---

## [Editor Report · Acceptance letter]

PONE-D-24-53437R1

PLOS ONE

Dear Dr. Bacal,

I'm pleased to inform you that your manuscript has been deemed suitable for publication in PLOS ONE. Congratulations! Your manuscript is now being handed over to our production team.

Kind regards,

on behalf of

Professor Qinghua Shi

Academic Editor

PLOS ONE